# SkyEvents: A Large-Scale Event-enhanced UAV Dataset for Robust 3D Scene Reconstruction

**Wenzong Ma**[1,*], **Zhuoxiao Li**[1,*], **Jinjing Zhu**[1,*], **Tongyan Hua**[1,*], **Kanghao Chen**[1], **Zidong Cao**[1],
**Da Yang**[2], **Peilun Shi**[3], **Yibo Zhou**[4], **Wufan Zhao**[1,†], **Hui Xiong**[1,†]

[1] Hong Kong University of Science and Technology (Guangzhou)
[2] Guangdong Airace Technology Co., Ltd.
[3] Chinese University of Hong Kong
[4] Beihang University

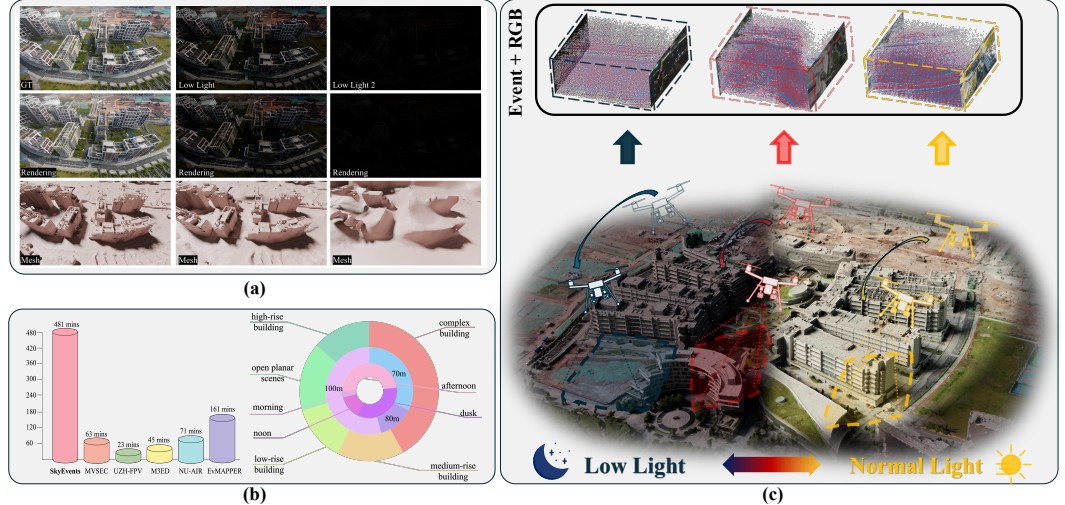

Figure 1: (a) Rendering and mesh under different light conditions. (b) Comparison and statistics of datasets. (c) Dataset collection across varied illumination conditions, scenarios, and flight altitudes.

## Abstract

Recent advances in large-scale 3D scene reconstruction using unmanned aerial vehicles (UAVs) have spurred increasing interest in neural rendering techniques. However, existing approaches with conventional cameras struggle to capture consistent multi-view images of scenes, particularly in extremely blurred and low-light environments, due to the inherent limitations in dynamic range caused by long exposure and motion blur resulting from camera motion. As a promising solution, bio-inspired event cameras exhibit robustness in extreme scenarios, due to their high dynamic range and microsecond-level temporal resolution. Nevertheless, dedicated event datasets specifically tailored for large-scale UAV 3D scene reconstruction remain limited. To bridge this gap, we introduce **SkyEvents**, a pioneering large-scale event-enhanced UAV dataset for 3D scene reconstruction, incorporating RGB, event, and LiDAR data. SkyEvents encompasses 45 sequences, spanning over 8 hours of video, captured across a diverse set of illumination conditions, scenarios, and flight altitudes. To facilitate the event-based 3D scene reconstruction with SkyEvents, we propose the Geometry-constrained Timestamp Alignment (**GTA**) module to align timestamps between the event and RGB cameras. Furthermore, we introduce a Region-wise Event Rendering (**RER**) loss for

---

*Equal contribution. † Corresponding authors: {wufanzhao, xionghui}@ust.hk.

supervising the rendering optimization. With SkyEvents, we aim to motivate and equip researchers to advance large-scale 3D scene reconstruction in challenging environments, harnessing the unique strengths of event cameras. Dataset and code will be available at `https://github.com/Anthony-ECPKN/SkyEvent`.

# 1 INTRODUCTION

Large-scale 3D scene acquisition using unmanned aerial vehicles (UAVs) has become a central tool for urban modeling, digital twins, and robotics applications. Driven by the growing demand for accurate and detailed reconstructions of city-scale environments, neural rendering techniques, such as Neural Radiance Fields (NeRF) (Mildenhall et al., 2021) and 3D Gaussian Splatting (3DGS) (Kerbl et al., 2023), have been extended from small object-centric captures (Barron et al., 2022; Müller et al., 2022; Li et al., 2023; Huang et al., 2024; Yu et al., 2024; Guédon & Lepetit, 2023) to large-scale scenes (Tancik et al., 2022; Turki et al., 2022; Mi & Xu, 2023; Lin et al., 2024; Yu Chen, 2024; Liu et al., 2025; Li et al., 2025). However, existing approaches still fundamentally rely on conventional CMOS-based RGB cameras, which remain highly susceptible to the limitations in challenging conditions-especially in scenarios with motion blur and insufficient illumination (Cladera et al., 2025). This inherent limitation often leads to a degradation in the quality of 3D reconstructions under low-light and motion-blurred conditions (Zahid et al., 2025; Matta et al., 2025; Zhang et al., 2025), thus limiting the achievable reconstruction fidelity (see Figure 1(a)).

To address this, event cameras have emerged as a promising alternative visual sensing paradigm, providing a complementary modality for 3D reconstruction. Unlike conventional cameras, event-based sensors asynchronously record changes in scene brightness at microsecond resolution, offering significantly improved temporal fidelity, a wide dynamic range, and robustness to motion blur (Xu et al., 2025; Zhu et al., 2025). These advantages have made event cameras increasingly popular in 3D reconstruction, where they have been successfully combined with NeRF and 3DGS techniques to enhance the robustness of reconstructions in dynamic and low-light environments (Yura et al., 2025; Zhang et al., 2025; Zhu et al., 2024).

Despite the growing efforts in event cameras for 3D reconstruction, a key limitation remains: *the lack of suitable, event-enhanced UAV datasets tailored for large-scale 3D scene reconstruction*. While recent event-enhanced UAV datasets have laid important groundwork, they often lack the necessary modalities and ground truth required for high-fidelity city-scale 3D reconstruction (in Table 1). For instance, datasets such as MVSEC (Zhu et al., 2018) and UZH-FPV (Delmerico et al., 2019) include aerial sequences, but they lack synchronized high-resolution RGB frames or volumetric ground truth, limiting their utility for neural rendering. Similarly, M3ED (Chaney et al., 2023) and NU-AIR (Iaboni et al., 2025) offer useful event streams and RGB imagery but do not provide the dense depth and 6-DoF pose supervision necessary for large-scale 3D reconstruction. The most recent work, EvMAPPER (Cladera et al., 2025), pioneers event-based orthomapping for high-altitude flights, but its focus on planar mosaics does not address the challenges of trajectory complexity and frame jitter that are inherent to low-altitude, large-scale 3D modeling.

To address it, we introduce *SkyEvents*, the first dataset specifically designed for event-enhanced UAV 3D reconstruction. SkyEvents brings together challenging low-light and motion-blurred conditions, synchronized RGB frames, dense per-frame depth supervision, high-quality 3D ground-truth reconstructions, and accurate 6-DoF UAV poses within a benchmark dataset. The data is collected using a DJI Matrice 350 equipped with a centimeter-accurate real-time kinematics system, flying over five distinct areas at altitudes ranging from 70 to 100 meters (see Figure 1(c)). DJI L2 LiDAR data serves as the ground truth due to its illumination-invariant nature and high precision. In total, the dataset contains 45 sequences (spanning over 8 hours) of paired RGB and event data, along with $0.72$ km$^2$ of point cloud data capturing at 2.64 cm/pixel GSD, enabling the robust development of perception algorithms under real-world conditions (see Figure 1(b)).

To enable event-based 3D reconstruction with SkyEvents, we provide two supporting components: **GTA** module, which aligns event and RGB data based on temporal constraints, and **RER** loss, which optimizes event-based rendering. We evaluate the effectiveness of the proposed GTA module and RER loss with existing neural rendering techniques using the SkyEvents dataset. Our experiments demonstrate the potential of event data for 3D scene reconstruction, particularly in challenging environments.

| Dataset | Low-light/night | RGB Rate | FoV (RGB/Event) | 3D Sup. (D/G/P)[1] | Resolution | Duration[2] |
|---|---|---|---|---|---|---|
| MVSEC (Hexacopter) (Zhu et al., 2018) | ✓[3] | ✗ | ✗/83° | D / ✗ / P | 346×260 | 63 |
| UZH-FPV (Delmerico et al., 2019) | ✗ | 30/50 Hz | 186°/120° | ✗ / ✗ / P | 346×260 | 23 |
| M3ED (UAV splits) (Chaney et al., 2023) | ✗ | 30 Hz | 52°/63° | D / ✗ / P | 1280×720 | 45 |
| NU-AIR (Iaboni et al., 2025) | ✗[4] | ✗ | ✗/70° | ✗ / ✗ / ✗ | 640×480 | 71 |
| EvMAPPER (Cladera et al., 2025) | ✓ | 50 Hz | 71°/64° | ✗ / ✗ / ✗ | 1280×720 | 161 |
| **SkyEvents** | **✓** | **120 Hz** | **71°/45°** | **D / G / P** | **1280×720** | **481** |

[1] 3D Sup. (D/G/P) denotes the availability of **D**epth, **G**eometry ground truth, and 6-DoF **P**ose. [2] In minutes.
[3] Dusk-only. [4] Multiple illumination conditions.

Table 1: Comparison of SkyEvents with previous event-based UAV datasets.

**Contributions.** (1) We present *SkyEvents*, a first large-scale event-enhanced UAV dataset for 3D scene reconstruction, including synchronized RGB and event data, LiDAR data, and accurate 6-DoF UAV poses. (2) We provide two supporting components for integrating event modality into neural rendering-based 3D reconstruction: GTA and RER modules. (3) Experiments demonstrate that event-guided neural rendering outperforms RGB-only baselines, achieving higher texture fidelity and geometric accuracy in large-scale 3D reconstructions under low-light and motion-blurred environments.

## 2 RELATED WORKS

### 2.1 3D SCENE RECONSTRUCTION

3D scene reconstruction seeks to recover the geometric structure of a scene from multi-view images or other modality data. Neural Radiance Fields (NeRF) (Deng et al., 2022; Garbin et al., 2021; Mildenhall et al., 2021; Hua & Wang, 2024) have demonstrated high-fidelity novel view synthesis through an implicit representation, but are constrained by slow optimization and limited geometric precision. In contrast, 3D Gaussian Splatting (3DGS) (Kerbl et al., 2023) leverages an explicit point-based representation to improve computational efficiency. Each Gaussian in 3DGS is characterized by parameters such as center, opacity, covariance, and color. Despite these improvements, the geometric accuracy and visual fidelity in 3DGS reconstructions can degrade under challenging conditions, such as motion-blurred or low-light environments (Zahid et al., 2025; Matta et al., 2025). To address these challenges, recent advancements in event-based 3DGS (Yura et al., 2025; Zhang et al., 2025) have shown promising results in enhancing reconstruction quality by utilizing event cameras. However, a notable gap in the existing works is the lack of relevant event datasets for large-scale 3D scene reconstruction. *To fill this gap, we introduce the first dataset specifically collected for event-enhanced UAV 3D scene reconstruction.*

### 2.2 EVENT-BASED 3D RECONSTRUCTION

Event cameras (Zheng et al., 2023; Xu et al., 2025) are bio-inspired sensors that capture asynchronous brightness changes, in contrast to traditional cameras, which capture images at a fixed frame rate. This unique sensing mechanism provides event cameras with distinct advantages, including exceptional temporal resolution, low latency, and resilience to motion blur and challenging lighting conditions. These attributes have spurred significant research into the application of event cameras across various computer vision tasks, such as object detection (Mitrokhin et al., 2018), depth estimation (Pan et al., 2024; Shi et al., 2023), semantic segmentation (Chen et al., 2024; Kong et al., 2024), video enhancement (Kim et al., 2024; Jing et al., 2021; Tulyakov et al., 2021), and, notably, 3D reconstruction (Chen et al., 2025; Wu et al., 2024; Cannici & Scaramuzza, 2024; Han et al., 2024; Cladera et al., 2025; Ye et al., 2025; Han et al., 2024; Yu et al., 2025). In the domain of 3D reconstruction, early methods (Rudnev et al., 2023; Zhu et al., 2024; Low & Lee, 2023) combined NeRF with event-based data, utilizing volumetric rendering techniques guided by event supervision. More recent works (Zahid et al., 2025; Matta et al., 2025; Yura et al., 2025; Zhang et al., 2025) have explored the integration of 3DGS with event data to enhance the reconstruction process. However, these prior approaches often face robustness issues, particularly in challenging conditions such as

low-light and motion-blurred environments. To address these shortcomings, Dark-EvGS (Wu et al., 2025) introduces an event-guided 3DGS pipeline, facilitating bright frame synthesis from arbitrary viewpoints in low-light scenarios. Although this approach represents a significant advancement, it remains primarily focused on small objects and does not fully capture the complexities of real-world 3D reconstruction tasks, which typically involve large-scale scenes. *In this work, we introduce a new dataset specifically designed for large-scale UAV 3D scene reconstruction, overcoming the limitations of previous methods in city-scale scenes.*

### 2.3 COMPARISON TO EXISTING EVENT-GUIDED DATASETS

Early UAV-related event datasets, such as MVSEC (Zhu et al., 2018) and UZH-FPV (Delmerico et al., 2019), established important baselines by providing stereo events with aggressive flight trajectories and precise ground truth. However, these datasets are not curated for texture-rich, city-scale 3D modeling (e.g., limited image resolution/coverage and a focus on odometry rather than volumetric ground truth). Subsequent multi-platform corpora like M3ED (Chaney et al., 2023) include aerial sequences and synchronized modalities, but primarily target high-speed robotics rather than low-altitude urban reconstruction with dense geometry. The NU-AIR dataset (Iaboni et al., 2025) advances urban perception with aerial event streams and extensive detection labels, while lacking synchronized high-resolution RGB and per-frame dense depth required for neural rendering benchmarks. EvMAPPER (Cladera et al., 2025) represents a pioneering effort in event-based orthomapping at high altitudes, yet it generates only planar mosaics and does not capture the pose jitter and parallax needed for volumetric modeling at low altitudes. Beyond datasets focused on UAVs, event camera datasets and benchmarks from other domains, including automotive driving (Gehrig et al., 2021; Binas et al., 2017) and indoor robotics (Fischer & Milford, 2020; Mitrokhin et al.; Burner et al., 2022), have advanced event vision, but they are not tailored to aerial 3D reconstruction at city scale because they lack synchronized RGB, dense per-frame depth, and low altitude capture protocols. *To bridge this gap, in this work, we introduce an event-enhanced UAV dataset tailored for large-scale 3D scene reconstruction.*

## 3 DATASET

We present **SkyEvents**, the first and large-scale UAV dataset that integrates tri-modal sensing across event camera streams, RGB videos, and LiDAR point clouds. This dataset spans five distinct environments, each characterized by unique architectural structures and diverse activity patterns. To address the critical challenge of accurate temporal synchronization across RGB and event data, we propose GTA module. The GTA module ensures precise alignment of timestamps between event and RGB cameras, leveraging geometric constraints to optimize synchronization (see Figure 2).

### 3.1 PLATFORM AND DATA COLLECTION

We build a UAV sensor rig using a DJI Matrice 350 RTK equipped with a Prophesee Gen4EVK event camera, a DJI Osmo Action 4 RGB camera, and an onboard mini-PC for synchronized logging (Figure 2; specs in Appendix Table 4). The mini-PC triggers both cameras under a shared time base and records activation timestamps for coarse synchronization. SkyEvents contains 45 sequences spanning over 8 hours across five areas (1.41 km$^2$), captured at 70–100 m altitude, together with a DJI L2 LiDAR survey (0.72 km$^2$) used as reference geometry and to derive dense depth. Per-sequence statistics and distributions are provided in Figure 3 and Appendix Table 5.

### 3.2 GEOMETRY-CONSTRAINED TIMESTAMP ALIGNMENT

In our system, we observe that the event camera stream is typically delayed by approximately 5 $ms$ relative to the RGB stream. Consequently, we need to ensure frame-accurate synchronization between event camera and RGB camera. Let $I_t \in \mathbb{R}^{H \times W \times 3}$ denote the RGB image at time $t$, and let $E_\tau \in \mathbb{R}^{H \times W \times 3}$ be the event-rendered image at time $\tau$. Given RGB sampling times $\{t_k\}_{k=1}^{K}$ (e.g., 1 s interval), we search a symmetric window $[t_k - \Delta, t_k + \Delta]$ with step $\delta$ and select the event time that maximizes a geometry score $\tau_k^\star$:

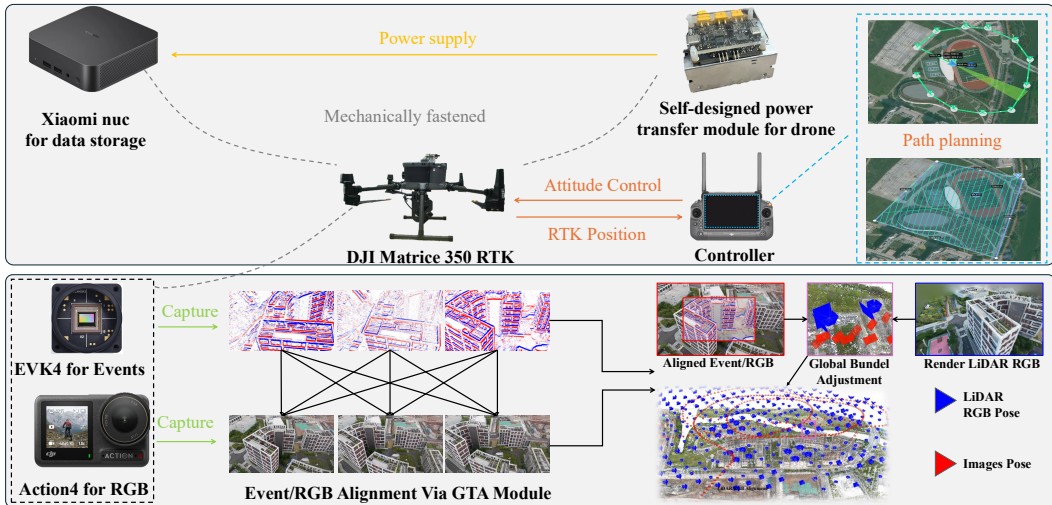

Figure 2: Data collection and rendering pipelines. The data acquisition platform consists of an UAV payload, an event camera, a 120HZ RGB camera, and a Mini PC. After collecting paired RGB and event data, we utilized the proposed GTA module to synchronize timestamps and warp between the event and RGB cameras.

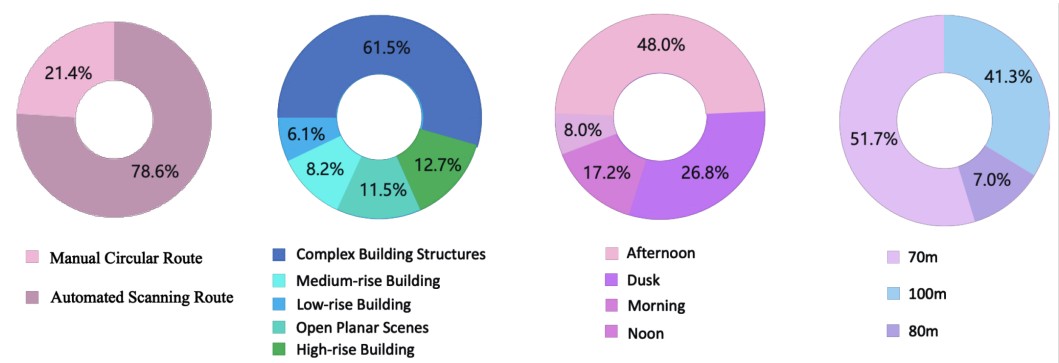

Figure 3: Data statistics from left to right: flight path, scenarios types, illumination, and height.

$$\tau_k^\star \in \arg\max_{\tau \in \mathcal{T}_k} S\big(I_{t_k}, E_\tau\big), \qquad \mathcal{T}_k = \{t_k - \Delta,\ t_k - \Delta + \delta, \ldots,\ t_k + \Delta\}. \tag{1}$$

where $\Delta = 100ms$ is the half-window size, $\delta = 8.333ms$ is the temporal step, $\mathcal{T}_k$ is the candidate set around $t_k$, and $S(\cdot, \cdot)$ is the geometry consistency score.

For a pair $(E_\tau, I_{t_k})$, we obtain putative correspondences $\{(\mathbf{x}_i, \mathbf{x}_i')\}_{i=1}^N$ with $\mathbf{x}_i = (u_i, v_i) \in E_\tau$ and $\mathbf{x}_i' = (u_i', v_i') \in I_{t_k}$ via the dense matcher MatchAnything/ROMA (He et al., 2025; Edstedt et al., 2024), and estimate a robust homography $\mathbf{H} \in \mathbb{R}^{3\times3}$ from RGB to event via MAGSAC (Barath et al., 2019):

$$\lambda \begin{bmatrix} u \\ v \\ 1 \end{bmatrix} = \mathbf{H} \begin{bmatrix} u' \\ v' \\ 1 \end{bmatrix}, \qquad \lambda \neq 0. \tag{2}$$

where $\mathbf{H}$ maps RGB pixels to Event pixels, $[u, v, 1]^\top$ and $[u', v', 1]^\top$ are homogeneous coordinates in Event and RGB, and $\lambda$ is the projective scale.

Let $\mathbf{m} \in \{0,1\}^N$ be the inlier mask and $\Pi([x,y,w]^\top) = (x/w, y/w)$. The per-inlier reprojection error is $\varepsilon_i = \|\Pi(\mathbf{H}[\mathbf{x}_i'; 1]) - \mathbf{x}_i\|_2$. We then score the pair as

$$S\big(I_{t_k}, E_\tau\big) = \sum_{i=1}^N m_i \; - \; \alpha \, \frac{\sum_{i=1}^N m_i \, \varepsilon_i}{\max\big(1, \sum_{i=1}^N m_i\big)}, \qquad \alpha > 0, \tag{3}$$

where $m_i$ indicates whether correspondence $i$ is an inlier, $\varepsilon_i$ is the Euclidean reprojection error for inlier $i$, and $\alpha$ balances inlier support and normalized error (invalid $\mathbf{H}$ yields a non-informative score).

To avoid per-pair perspective warping while preserving alignment, we approximate the RGB→Event homography $\mathbf{H}$ on a regular grid by a diagonal affine map $[x,y]^\top \approx \mathbf{D}[x',y']^\top + \mathbf{t}$ with $\mathbf{D} = \mathrm{diag}(s_x, s_y)$ and $\mathbf{t} = (t_x, t_y)^\top$, and estimate $(s_x, s_y, t_x, t_y)$ via a *single linear least-squares fit*. Concretely, we sample $M$ grid points $(x_j', y_j')$ in RGB, project them to Event by $(x_j, y_j) = \Pi\big(\mathbf{H}[x_j', y_j', 1]^\top\big)$, and solve $\min_{\boldsymbol{\theta}} \|\mathbf{A}\boldsymbol{\theta} - \mathbf{b}\|_2^2$, where the unknown $\boldsymbol{\theta} = (s_x, t_x, s_y, t_y)^\top$, matrix $\mathbf{A}$ stacks RGB coordinates with axis-wise structure, and $\mathbf{b}$ stacks the corresponding Event coordinates (first all $x_j$, then all $y_j$). Given event and RGB resolutions $(W_0, H_0)$ and $(W_1, H_1)$, we derive an RGB crop window $(x_0, y_0) \to (x_1, y_1)$ by stabilizing scales with $\tilde{s}_x = \max(s_x, \epsilon)$, $\tilde{s}_y = \max(s_y, \epsilon)$, back-computing the crop origin from $(-t_x/\tilde{s}_x, -t_y/\tilde{s}_y)$ with clamping to valid bounds, and setting the crop size so that a bilinear resize matches the event resolution, i.e., widths/heights proportional to $W_0/\tilde{s}_x$ and $H_0/\tilde{s}_y$. We log $\mathbf{H}, \boldsymbol{\theta}$, and the crop coordinates for exact reproducibility, where $\epsilon > 0$ is a small stabilizer.

To enforce a global $1\,\mathrm{second}$ cadence and suppress local noise, we jointly refine the sequence by maximizing geometric consistency penalized by interval deviations:

$$\{\widetilde{\tau}_k\}_{k=1}^K = \arg\max_{\{\tau_k\}} \left[ \sum_{k=1}^K S\big(I_{t_k}, E_{\tau_k}\big) - \beta \sum_{k=2}^K \big|(\tau_k - \tau_{k-1}) - 1\,\mathrm{s}\big| \right], \qquad \beta > 0. \tag{4}$$

where $\widetilde{\tau}_k$ are the refined event timestamps; $\beta$ trades off the global $1\,\mathrm{second}$ cadence against geometric fit, and the absolute deviation term penalizes interval drift.

## 3.3 LiDAR Alignment

As demonstrated in Figure 2, we align LiDAR and RGB in a unified 3D coordinate and then transfer LiDAR geometry to each RGB image as metrically accurate depth. In our setup, a DJI Zenmuse L2 LiDAR payload and an RGB camera are flown in separate missions over the same area, so the raw trajectories are not temporally synchronized. To recover a consistent geometry, we adopt a unified structure-from-motion (SfM) pipeline using commercial photogrammetry software *RealityScan*. We first rasterize the LiDAR point cloud into perspective "LiDAR RGB" images by projecting the laser returns onto virtual pinhole cameras whose intrinsics and extrinsics are derived from the LiDAR trajectory. RealityScan generates the poses of these LiDAR images, and we treat them as fixed anchor views. We then import the calibrated UAV RGB images and run joint SfM and global bundle adjustment over all images, keeping the LiDAR-derived camera poses frozen. This procedure rigidly registers all RGB cameras into the LiDAR coordinate system, yielding a single globally consistent Euclidean frame shared by LiDAR and RGB.

Given the optimized camera intrinsics and extrinsics, we perform a global multi-view stereo (MVS) reconstruction in which LiDAR depth acts as a strong geometric prior. Specifically, we fuse the LiDAR point cloud with MVS-derived depth estimates to obtain a dense, continuous mesh, where LiDAR stabilizes depth in texture-poor or repetitive regions and suppresses multi-view ambiguities. We treat this fused surface as the ground-truth (GT) geometry. Finally, for each RGB frame, we back-project the GT mesh into the corresponding camera using the calibrated intrinsics and extrinsics, producing dense, metrically accurate depth maps that are pixel-wise aligned with the original RGB images.

| Scenes | Conditions | Methods | With Event | | | Without Event | | |
|--------|-----------|---------|------------|---|---|---------------|---|---|
| | | | SSIM↑ | PSNR↑ | LPIPS↓ | SSIM↑ | PSNR↑ | LPIPS↓ |
| Scene1 | Low-light | Luminance-GS | 0.1257 | 5.2060 | 0.5850 | 0.1214 | 4.7870 | 0.5990 |
| | Blur | Improved-GS | 0.8044 | 27.4368 | 0.2701 | 0.8095 | 27.3554 | 0.2757 |
| | | Improved-GS+kernel | 0.8625 | 28.2600 | 0.2107 | 0.8655 | 28.1146 | 0.2045 |
| Scene2 | Low-light | Luminance-GS | 0.1417 | 5.7650 | 0.5402 | 0.1408 | 5.7000 | 0.5411 |
| | Blur | Improved-GS | 0.7951 | 26.4789 | 0.2482 | 0.7807 | 25.8635 | 0.2653 |
| | | Improved-GS+kernel | 0.2608 | 11.5674 | 0.6783 | 0.2488 | 11.5092 | 0.6816 |

Table 2: Rendering performance comparison across different conditions and scenarios, with and without event data.

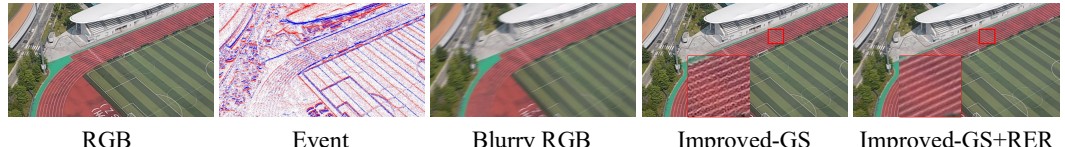

RGB      Event      Blurry RGB      Improved-GS      Improved-GS+RER

Figure 4: Comparison of 3D scene reconstruction with Improved-GS and Improved-GS+RER in blurred environments.

## 4 BENCHMARKS

### 4.1 BENCHMARK METHOD

To evaluate the effectiveness of incorporating event modality for 3D reconstruction, we leverage two state-of-the-art 3DGS pipelines, Luminance-GS (Cui et al., 2025) and Improved-GS (Deng et al., 2025), for benchmark experiments.

**Input.** Each event $e_i = (\mathbf{x}, t_i, p_i)$ is triggered at a microsecond timestamp $t_i$ when the brightness at pixel $\mathbf{x}$ changes by more than a contrast threshold $C$ (i.e., $|L(\mathbf{x}, t_{i+1}) - L(\mathbf{x}, t_i)| \geq C$), where $L(\mathbf{x}, t_i)$ is the logarithmic brightness, and $p_i \in \{-1, +1\}$ represents either an increase or decrease in the logarithmic brightness $L(\mathbf{x}, t_i)$. In parallel, RGB image $I_{t_i}$ is captured at discrete time $t_i$.

### 4.2 REGION-WISE EVENT RENDERING LOSS

To encourage 3DGS to recover accurate geometry and appearance, we introduce an event-based brightness-change consistency loss Zhang et al. (2025). Given two timestamps $t_1$ and $t_2$, we accumulate events within $(t_1, t_2)$ into an event image

$$\bar{E}(t_1, t_2)(\mathbf{x}) = \sum_{t_1 < t_i < t_2} p_i \, \mathbf{1}[\mathbf{x}_i = \mathbf{x}], \tag{5}$$

which approximates the log-brightness change over the interval. We compare it with the synthesized brightness change, computed as the logarithmic difference between two rendered images $\hat{I}_{t_1}$ and $\hat{I}_{t_2}$. Since the RGB and event sensors have different footprints and intrinsics, naively warping and undistorting RGB images leads to misalignment with the event frame. Instead, we estimate the warp between the two sensors and define a region-aligned event supervision loss that constrains the brightness change only within their overlapping regions. To ensure spatial support is consistent with the event frame, we apply the region-wise alignment and cropping derived in equation 1: we approximate the RGB→event homography by a diagonal affine map $\mathcal{C}_{\boldsymbol{\theta}}$ with parameters $\boldsymbol{\theta} = (s_x, s_y, t_x, t_y)$, derive a crop window, and resample to the event resolution. We then convert the aligned renderings to log space and define the loss as:

$$\mathcal{L}_{\text{event}} = \left\| \left( \log \mathcal{C}_{\boldsymbol{\theta}}(\hat{I}_{t_2}) - \log \mathcal{C}_{\boldsymbol{\theta}}(\hat{I}_{t_1}) \right) - \bar{E}(t_1, t_2) \right\|_2^2. \tag{6}$$

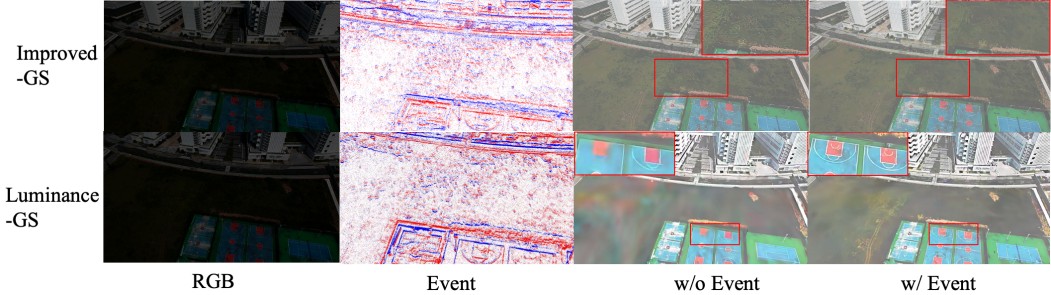

Figure 5: Comparison of 3D scene reconstruction using existing 3D GS methods (Improved-GS and Luminance-GS), with and without event enhancement. The integration of event modality through RER markedly enhances rendering quality.

### 4.3 IMPLEMENTATION DETAILS AND EVALUATION METRICS

**Implementation Details.** For the training setup, we employ Luminance-GS (Cui et al., 2025), a state-of-the-art (SOTA) 3DGS pipeline designed for complex illumination scenes, and Improved-GS (Deng et al., 2025), the SOTA neural rendering pipeline, to assess the effectiveness of incorporating event modality. Both models were trained on a single NVIDIA RTX 4090 using the Adam optimizer. Training is conducted for a total of 30,000 iterations, with Event refinement starting at 8,000 iterations. For other settings, such as Gaussian reset steps (Kerbl et al., 2023), we follow the default configuration.

**Low-light Image Generation.** Although UAVs can be programmed to repeat nearly identical RTK-guided flight paths across different light conditions, to build a unified model, we rely on structure-from-motion (SfM), which requires reliable feature detection and matching across views. Under low-light conditions, however, many images exhibit weak textures and low contrast, causing feature matching to fail. As a result, a substantial portion of low-light images cannot be registered in the SfM pipeline, preventing consistent multi-temporal 3D reconstruction. To evaluate performance of methods in low light-conditions, we utilize daytime sequences in our experiments and generate synthetic low-light versions. This ensures pixel-level correspondence, enabling controlled training and ablation studies. Following (Zhang et al., 2021; Liang et al., 2023), we generate low-light frames from corresponding normal-light frames using gamma correction and linear scaling, employing identical parameter settings. The process is formalized as follows: $L_t(p) = \beta \times (\alpha \times I_t(p))^\gamma$, where $\gamma$ represents the gamma correction factor, which is sampled from a uniform distribution U(2, 3.5). The parameters $\alpha$ and $\beta$ are linear scaling factors, drawn from U(0.9, 1) and U(0.5, 1), respectively.

**Blurry Image Generation.** Following prior work (Li et al., 2024), a motion-blurred image is physically generated by accumulating photons over the exposure time, ensuring that the resulting blurred image remains differentiable with respect to both the parameters of NeRF and the motion trajectory.

**Evaluation Metrics.** To thoroughly evaluate the performance of 3DGS methods, we employ standard quantitative metrics, including Peak Signal-to-Noise Ratio (PSNR), Structural Similarity Index Measure (SSIM), and Learned Perceptual Image Patch Similarity (LPIPS) (Zhang et al., 2018).

### 4.4 RESULTS

In this section we focus on novel view synthesis from joint RGB and event input using three dimensional Gaussian splatting backbones. The goal is to answer a concrete question: *given a fixed rendering pipeline, does adding events improve reconstruction quality in low light and motion blurred UAV scenes?*

Table 2 summarizes the quantitative results on two representative scenes. For the low light setting with Luminance GS (Cui et al., 2025), events provide small but consistent gains. On both scenes, adding event supervision increases PSNR and slightly reduces LPIPS, while keeping SSIM essentially unchanged, which indicates that events act as a stabilizing cue under extreme illumination where RGB alone is severely underexposed. For the blurred setting with Improved GS (Deng et al.,

| Method | BRISQUE ↓ | MANIQA ↑ | NIQE ↓ |
|---|---|---|---|
| E2VID | **7.1075** | 0.2141 | 6.1096 |
| E2VID+ | 7.3820 | 0.2842 | 5.4571 |
| ET-Net | 22.9103 | 0.2885 | 5.4518 |
| FireNet | 14.3349 | 0.2276 | 4.5584 |
| FireNet+ | 12.7435 | **0.3445** | **4.2509** |
| HyperE2VID | 7.6182 | 0.2205 | 5.7398 |
| SPADE-E2VID | 14.7488 | 0.2444 | 7.0037 |
| SSL-E2VID | 59.9326 | 0.1701 | 9.0612 |

Table 3: Quantitative comparison of event-to-video methods.

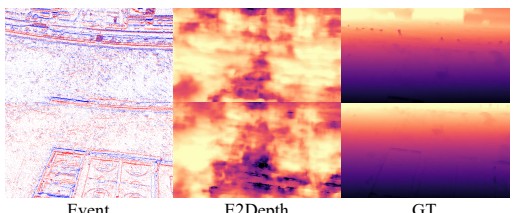

Event      E2Depth      GT

Figure 6: Depth Estimation.

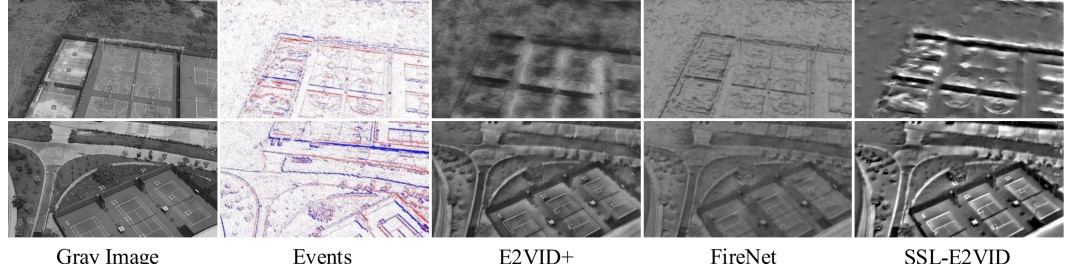

Gray Image      Events      E2VID+      FireNet      SSL-E2VID

Figure 7: Video reconstruction performance comparison: E2VID+ (Stoffregen et al., 2020), FireNet (Scheerlinck et al., 2020), and SSL-E2VID (Paredes-Vallés & De Croon, 2021).

2025), the benefit of events is more pronounced. On the larger Scene 2 with more than 800 training images, the gains are clearer: PSNR increases by about half a decibel and LPIPS decreases by a noticeable margin for the event driven model compared to the RGB only baseline. These trends hold both for the plain Improved GS variant and for the simple gamma recovery kernel, which shows that event cues remain helpful even when exposure is partially corrected on the RGB branch. Figure 4 visualizes the baseline comparison that underlies the blurred rows for Improved GS in Table 2. When events are used, double contours and ghosting around moving or blurred regions are notably reduced and details appear sharper, which matches the quantitative improvements in PSNR and LPIPS. Under normal lighting, minor camera jitters that cause ambiguities in the RGB only reconstructions are also mitigated once event constraints are introduced.

Overall, these results suggest that events supply high-frequency constraints that particularly aid deblurring, and also benefit low-light reconstruction, especially in challenging or large-scale settings.

### 4.5 OTHER TASKS TO EXPLORE

**Monocular Depth Estimation.** We further evaluate event-based monocular depth estimation on SkyEvents by running inference with the SOTA E2Depth model. As illustrated in Figure 6, depth predictions exhibit noticeable artifacts and loss of fine structures in aerial event streams, highlighting that current models do not adequately address UAV-based event depth estimation and underlining the potential value of our dataset for this frontier task.

**Event-to-Video Reconstruction.** Table 3 and Figure 7 summarize the performance of event-to-video reconstruction methods on our SkyEvents. Overall, methods originally trained on ground-level data struggle to generalize to UAV-based event streams. Methods such as ET-Net, SPADE-E2VID, and SSL-E2VID exhibit significantly degraded image quality, underscoring the challenge posed by our aerial, low-light event data and the need for dedicated training in this regime.

## 5 CONCLUSION

We introduce SkyEvents, the first large-scale, event-enhanced UAV dataset specifically designed for 3D scene reconstruction using RGB, event, and LiDAR data. The dataset spans a wide range of conditions, including variations in illumination, scenario, and flight height, comprising 45 sequences (>8h) of paired RGB and event data, along with 0.72 km$^2$ of LiDAR point cloud data capturing at

around 2.64 cm/pixel GSD. To fully harness the potential of this large-scale dataset, we propose the GTA module to effectively synchronize RGB-event timestamps, and the RER loss to guide rendering optimization. Experimental results underscore the significant contribution of event data to 3D scene reconstruction. In conclusion, we hope SkyEvents will catalyze further research and innovation in the domain of event-enhanced 3D reconstruction, particularly in extreme scenarios.

**Limitation and Future Work**. Due to current hardware constraints, the RGB and event data are not perfectly aligned spatiotemporally. In the future, we aim to upgrade the sensor platform to obtain a larger, more precise, and higher-quality paired dataset, enabling more accurate synchronization under increasingly challenging scenarios. This enhanced dataset will serve as a foundation for exploring a broader range of computer vision tasks, such as depth estimation and video frame interpolation.

ACKNOWLEDGMENTS

This work was supported in part by the National Key R&D Program of China (Grant No. 2023YFF0725001), in part by the National Natural Science Foundation of China (Grant No. 92370204 and No. 42401567), in part by the Guangdong Basic and Applied Basic Research Foundation (Grant No. 2023B1515120057), in part by the Key-Area Special Project of Guangdong Provincial Ordinary Universities (Grant No. 2024ZDZX1007), in part by the AI Research and Learning Base of Urban Culture, Guangdong Provincial Department of Education (Grant No. 2023WZJD008), and in part by the Guangdong Provincial Project (Grant No. 2024QN11G095).

The authors would like to thank Guangdong Airace Technology Co., Ltd. and All-Airspace Intelligent Lab for their support and providing the experimental environment for this research.

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

# A APPENDIX

## A.1 UAV PLATFORM & SENSOR SYSTEM DESIGN

The data acquisition system was built around a DJI Matrice 350 RTK unmanned aerial vehicle, selected for its robust payload capacity of 2.7 kg and extended flight endurance of approximately 45 minutes, shown in Figure 8. This platform integrates RTK GNSS positioning with centimeter-level accuracy, ensuring precise georeferencing throughout all missions. The airframe's IP55 weather resistance enables operations under varying environmental conditions.

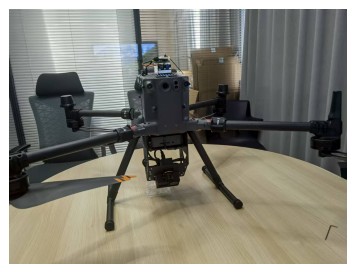 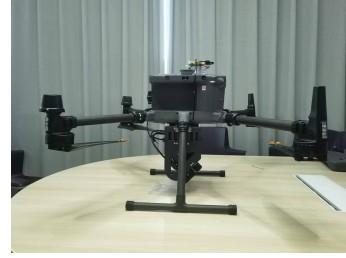 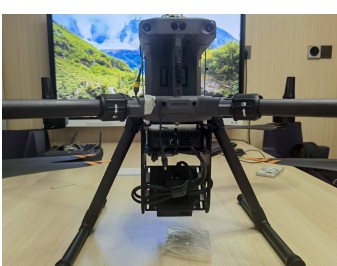

(a) Front view      (b) Side view      (c) Rear view

Figure 8: Multi-modal UAV data collection platform overview. The DJI Matrice 350 RTK is equipped with a synchronized sensor suite and An onboard mini-PC.

A custom-designed sensor suite was mounted on a vibration-damped carbon fiber plate to minimize motion artifacts, shown in Figure 9. The core vision system consists of two synchronized cameras: a Prophesee EVK4 HD event camera capturing asynchronous events at 1280×720 resolution with sub-millisecond latency, and a DJI Osmo Action 4 RGB camera recording 4K video at 120 fps with global shutter. Temporal alignment between these sensors was maintained with microsecond-level precision.

## A.2 DATA COLLECTION SYSTEM KEY COMPONENTS

An onboard Xiaomi Mini PC served as the central computing unit, handling real-time data acquisition from both cameras through USB 3.2 interfaces. With sustained write speeds exceeding 500 MB/s, this system reliably captured the high-bandwidth event streams (up to 10 Mevts/s) alongside uncompressed RGB video. The complete setup represents a balanced integration of commercial components and custom mounting solutions, providing a reproducible platform for multi-modal aerial data collection across diverse urban environments. Platform detail are listed in Table 4

## A.3 LARGE-SCALE SCENARIO DIVERSITY

The dataset encompasses five distinct scenes at Hong Kong University of Science and Technology (Guangzhou), shown in Figure 10, captured in sequential order: **(1) Main Teaching Building (2) North Dormitory, (3) Data Center, (4) Sports Field, (5) South Dormitory**. Details are listed in Figure 10 and Table 1 and each scene presents unique architectural and environmental challenges for aerial perception:

- **Main Teaching Complex**: A vast interconnected structure comprising ∼10 buildings with glass facades, featuring a central aerial garden with dense foliage, ground-level water bodies, and tree clusters.
- **North/South Dormitories**: High-rise residential buildings with rich texture patterns. The North Dormitory was sampled at around 70m, while the taller South Dormitory required 100m flight altitude for comprehensive coverage. Both feature intricate shadow dynamics from vertical structures.
- **Data Center**: Three isolated white-toned buildings with minimal color variation, creating texture-poor surfaces ideal for testing feature extraction algorithms. Uniform 70m sampling height.

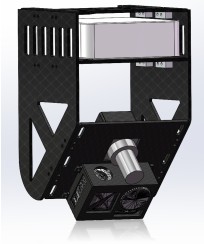
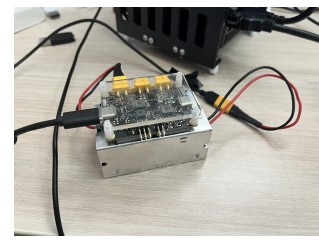
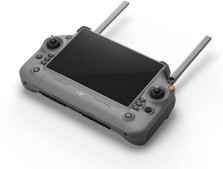

(a) Carbon fiber mounting plate

(b) Power conversion module

(c) Flight controller with route planning

Figure 9: Key components of the UAV data collection system. (a) Custom carbon fiber plate provides rigid mounting for sensors and computing unit while damping vibrations. (b) Power conversion module regulates DJI battery output to stable 12V/5V for onboard electronics. (c) Remote controller with pre-loaded KML routes enables fully autonomous flight operations.

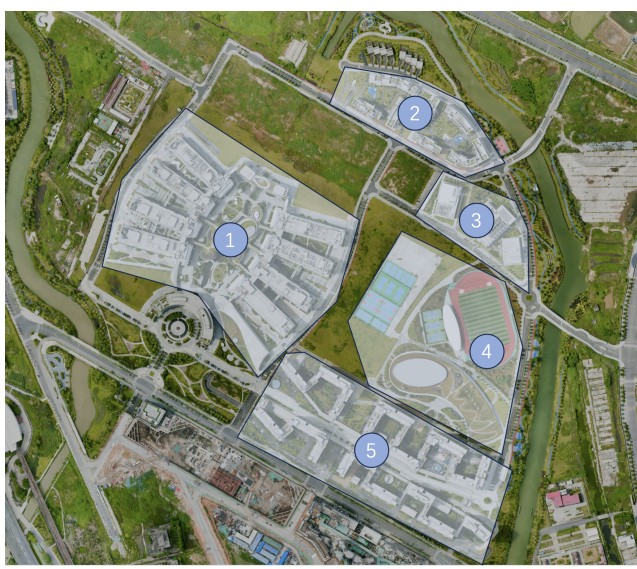

Figure 10: Campus region overview. The map shows five distinct areas: (1) Main Building complex, (2) North Dormitory, (3) Data Center, (4) Playground, and (5) South Dormitory, covering diverse urban scenarios for multi-modal data collection.

- **Sports Field**: Complementary low-texture environments characterized by monochromatic surfaces. The outdoor Sports Field features uniform turf patterns, while the indoor Gymnasium contains fast-moving human activities under variable artificial lighting.

### A.4 MATCHING RESULTS

Figure 11 shows the matching results with the proposed GTA module on SkyEvents and MVSEC datasets. And as shown in the figure, the event streams aligned by our GTA module exhibit a pixel-perfect correspondence with the RGB frames' geometric edges. Moreover, the successful alignment on MVSEC confirms that our module generalizes well and achieves high temporal precision.

### A.5 OTHER TASKS TO EXPLORE

Figure 13 shows the performance of existing event-to-video reconstruction methods E2VID (Rebecq et al., 2019), E2VID+ (Stoffregen et al., 2020), FireNet (Scheerlinck et al., 2020), ET-Net (Weng et al., 2021), FireNet+ (Stoffregen et al., 2020), HyperE2VID (Ercan et al., 2024),

Table 4: Precise Sensor Suite Specifications and Integration Details

**DJI Matrice 350 RTK (UAV Platform)**

| Category | Specification / Details |
| --- | --- |
| Role | Primary aerial vehicle |
| Dimensions | **Folded**: 430 × 420 × 430 mm (with props) |
| | **Unfolded**: 810 × 670 × 430 mm |
| Weight | **Without battery**: 3.77 kg; **With dual battery**: 6.47 kg |
| Payload | **Capacity**: 2.7 kg (max takeoff weight 9.2 kg); **Gimbal load**: 960 g max |
| Flight | **Endurance**: ∼55 min (unloaded, 8 m/s cruise) |
| | **Max speed**: 23 m/s (horizontal), 6 m/s (ascent), 5 m/s (descent) |
| | **Rotation rate**: Pitch: 300°/s, Yaw: 100°/s |
| Positioning | **System**: RTK GNSS (GPS+GLONASS+BeiDou+Galileo); **Accuracy**: 1 cm +1 ppm (horiz), 1.5 cm +1 ppm (vert) |
| Environmental | **Rating**: IP55 (weather resistant); **Max altitude**: 5000 m (with 2110s props) |
| Safety | ADS-B receiver, dual battery redundancy |
| Integration | Custom vibration-damped carbon fiber plate |

**Prophesee EVK4 HD (Event Camera)**

| Category | Specification / Details |
| --- | --- |
| Role | Asynchronous event capture |
| Resolution | 1280 × 720 (HD) |
| Pixel size | 4.86 × 4.86 $\mu$m |
| Temporal | **Latency**: 220 $\mu$s; **Event Rate**: 10 Mevts/s (million events per second) |
| Dynamic Range | 86 dB (up to 120 dB under low light) |
| Spectral | **Response**: 400-1000 nm (visible to NIR) |
| Power | **Consumption**: 0.5 W (USB powered) |
| Interface | **Data**: USB 3.0 Type-C; **Sync**: IX Connector Type B (sync in/out, trigger in) |
| Mechanical | **Dimensions**: 30 × 30 × 36 mm; **Weight**: 40 g (excluding lens) |
| Mounting | Rigid co-location with RGB camera |
| Optics | **Included**: C-mount 1/2.5" lens (FOV 47.7°) |
| Calibration | Temporal-spatial calibration with RGB |

**DJI Osmo Action 4 (RGB Camera)**

| Category | Specification / Details |
| --- | --- |
| Role | Synchronized RGB video capture |
| Resolution | 4K UHD (3840 × 2160) @ 120 fps |
| Shutter | Global shutter (eliminates rolling shutter artifacts) |
| Sensor | **Size**: 1/1.3" CMOS; **Pixel Size**: 2.4 $\mu$m |
| Dynamic Range | D-Log M (10-bit color depth) |
| Lens | **FOV**: 155° (wide mode) |
| Synchronization | **Accuracy**: <100 $\mu$s relative to event camera |

**Xiaomi Mini PC (Compute & Data Logger)**

| Category | Specification / Details |
| --- | --- |
| Role | Central data acquisition hub |
| CPU | Intel N100 (4-core, 3.4GHz Turbo) |
| Memory | 16GB DDR4 RAM |
| Storage | 512GB NVMe SSD |
| I/O | **Ports**: 4 × USB 3.2 Gen 2 (10Gbps), 1 × HDMI 2.0, 1 × 2.5G Ethernet |
| OS | Windows 10 Professional |
| Power | **Consumption**: 12W TDP (powered via UAV battery) |
| Data Rate | **Capacity**: ∼500 MB/s sustained write |

SPADE-E2VID (Cadena et al., 2021), and SSL-E2VID (Paredes-Vallés & De Croon, 2021) when applied to our SkyEvents dataset.

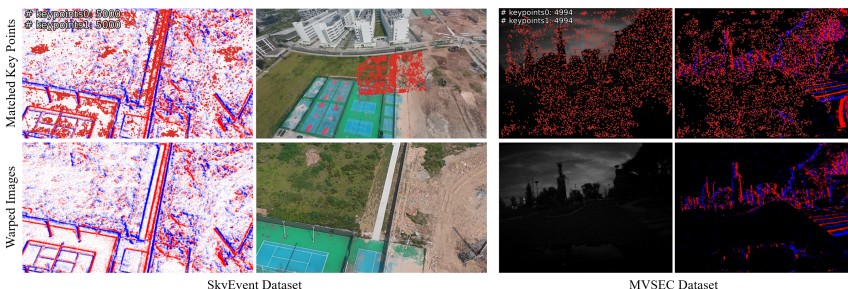

Figure 11: Matching results with the proposed GTA module on SkyEvents and MVSEC datasets.

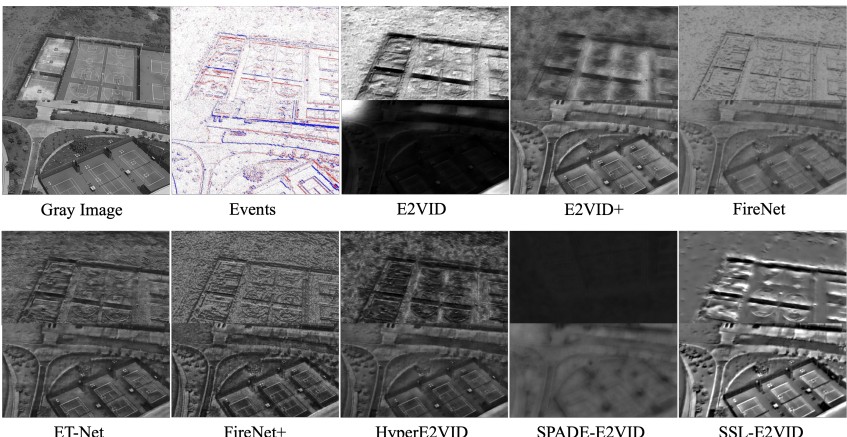

Figure 12: Video reconstruction with E2VID, E2VID+, FireNet, ET-Net, FireNet+, HyperE2VID, SPADE-E2VID, and SSL-E2VID.

Table 5: SkyEvents Dataset Collection Statistics

| Sequence | Sensor | Illumin. | Dur.(s) | Resolution | Scenario | Flight Path | Height(m) | Data Size(GB) |
|---|---|---|---|---|---|---|---|---|
| 07061519 | RGB | Afternoon | 1056 | 3840×2160 | Medium-rise building | Auto flight | 70 | 16.00 |
| 07061519 | Event | Afternoon | 1435 | 1280×720 | Medium-rise building | Auto flight | 70 | 115.00 |
| 07061537 | RGB | Afternoon | 379 | 3840×2160 | Medium-rise building | Auto flight | 70 | 5.54 |
| 07061635 | RGB | Afternoon | 1056 | 3840×2160 | Low-rise building | Auto flight | 70 | 16.00 |
| 07061635 | Event | Afternoon | 1056 | 1280×720 | Low-rise building | Auto flight | 70 | 141.00 |
| 07071011 | RGB | Morning | 1056 | 3840×2160 | Open planar scenes | Auto flight | 70 | 16.00 |
| 07071011 | Event | Morning | 1523 | 1280×720 | Open planar scenes | Auto flight | 70 | 103.00 |
| 07071029 | RGB | Morning | 467 | 3840×2160 | Open planar scenes | Auto flight | 70 | 7.04 |
| 07071146 | RGB | Noon | 1056 | 3840×2160 | High-rise building | Auto flight | 100 | 16.00 |
| 07071146 | Event | Morning | 1820 | 1280×720 | High-rise building | Auto flight | 100 | 150.00 |
| 07071204 | RGB | Noon | 764 | 3840×2160 | High-rise building | Auto flight | 100 | 11.50 |
| 07071500 | RGB | Afternoon | 1056 | 3840×2160 | Complex building | Auto flight | 100 | 16.00 |
| 07071500 | Event | Afternoon | 1628 | 1280×720 | Complex building | Auto flight | 100 | 145.00 |
| 07071518 | RGB | Afternoon | 572 | 3840×2160 | Complex building | Auto flight | 100 | 8.68 |
| 07071631 | Event | Afternoon | 1618 | 1280×720 | Complex building | Auto flight | 100 | 139.00 |
| 07071632 | RGB | Afternoon | 1056 | 3840×2160 | Complex building | Auto flight | 100 | 16.00 |
| 07071649 | RGB | Afternoon | 562 | 3840×2160 | Complex building | Auto flight | 100 | 8.47 |
| 07071800 | RGB | Dusk | 1056 | 3840×2160 | Low-rise building | Auto flight | 70 | 16.00 |
| 07071800 | Event | Dusk | 1183 | 1280×720 | Low-rise building | Auto flight | 70 | 95.70 |
| 07071817 | RGB | Dusk | 127 | 3840×2160 | Open planar scenes | Auto flight | 70 | 1.92 |
| 07071936 | RGB | Dusk | 1454 | 3840×2160 | Medium-rise building | Auto flight | 70 | 13.40 |
| 07071936 | Event | Dusk | 1454 | 1280×720 | Medium-rise building | Auto flight | 70 | 40.20 |
| 09111836 | RGB | Dusk | 546 | 3840×2160 | Complex building | Manual circle | 70 | 8.21 |
| 09111836 | Event | Dusk | 546 | 1280×720 | Complex building | Manual circle | 70 | 7.89 |
| 09121548 | RGB | Afternoon | 969 | 3840×2160 | Open planar scenes | Manual circle | 80 | 14.50 |
| 09121549 | Event | Afternoon | 969 | 1280×720 | Open planar scenes | Manual circle | 80 | 12.80 |
| 09121611 | RGB | Afternoon | 911 | 3840×2160 | Open planar scenes | Auto flight | 80 | 13.60 |
| 09121612 | Event | Afternoon | 911 | 1280×720 | Open planar scenes | Auto flight | 80 | 30.40 |
| 09141514 | RGB | Afternoon | 1053 | 3840×2160 | Complex building | Auto flight | 100 | 16.00 |
| 09141514 | Event | Afternoon | 1913 | 1280×720 | Complex building | Auto flight | 100 | 72.50 |
| 09141532 | RGB | Afternoon | 860 | 3840×2160 | Complex building | Auto flight | 100 | 12.40 |
| 09141551 | RGB | Afternoon | 752 | 3840×2160 | Complex building | Auto flight | 100 | 10.80 |
| 09141551 | Event | Afternoon | 752 | 1280×720 | Complex building | Auto flight | 100 | 23.80 |
| 09141605 | RGB | Afternoon | 1066 | 3840×2160 | Complex building | Manual circle | 80 | 15.90 |
| 09141605 | Event | Afternoon | 1066 | 1280×720 | Complex building | Manual circle | 80 | 56.30 |
| 09141759 | RGB | Dusk | 1053 | 3840×2160 | Complex building | Auto flight | 100 | 16.00 |
| 09141759 | Event | Dusk | 1803 | 1280×720 | Complex building | Auto flight | 100 | 104.00 |
| 09141816 | RGB | Dusk | 750 | 3840×2160 | Complex building | Auto flight | 100 | 10.90 |
| 09191513 | RGB | Afternoon | 782 | 3840×2160 | Low-rise building | Manual circle | 100 | 11.30 |
| 09191513 | Event | Afternoon | 782 | 1280×720 | Low-rise building | Manual circle | 100 | 22.80 |
| 09191530 | RGB | Afternoon | 599 | 3840×2160 | Medium-rise building | Manual circle | 100 | 8.76 |
| 09191530 | Event | Afternoon | 599 | 1280×720 | Medium-rise building | Manual circle | 100 | 24.10 |
| 09191543 | RGB | Afternoon | 1053 | 3840×2160 | High-rise building | Manual circle | 100 | 16.00 |
| 09191543 | Event | Afternoon | 1208 | 1280×720 | High-rise building | Manual circle | 100 | 52.20 |
| 09191601 | RGB | Afternoon | 155 | 3840×2160 | High-rise building | Manual circle | 100 | 2.08 |
| 0922 | LiDAR | - | 6600 | - | All regions | Auto flight | 100 | 362.00 |
| **Total Duration & Size** | | | | | | | **28866s** | **2022.69GB** |

[*] RGB: 3840×2160@30fps H.265; Event: 1280×720 @ variable rate; LiDAR: Full area coverage

## A.6 DATASET STATISTICS AND EXAMPLES

### A.6.1 DATASET STATISTICS

The SkyEvents dataset comprises multi-modal aerial data collected across five distinct urban scenarios, including complex building structures, low-rise and high-rise buildings, and open planar scenes. Data acquisition was performed using a UAV platform equipped with synchronized RGB (3840×2160), event (1280×720), and LiDAR sensors. Flights were conducted at altitudes of 70–100 m under varied illumination conditions (morning, noon, afternoon, dusk), with both automated routes and manual circular paths employed to ensure comprehensive spatial coverage. The dataset spans a total duration of 28,866 seconds, amounting to 2.02 TB of data. Detailed specifications for each sequence—including sensor types, illumination, resolution, flight parameters, and data size—are summarized in Table 5, where the exact values should be determined by the files released in the official download links. Minor discrepancies may exist due to file-system conventions, compression metadata, or rounding. In addition, DJI Action 4 automatically splits long video recordings into multiple files during acquisition. Therefore, the number of RGB video segments can be larger than the number of event recordings, and one event recording may correspond to multiple RGB video segments. This diverse collection supports development and evaluation of perception algorithms under real-world urban scenarios.

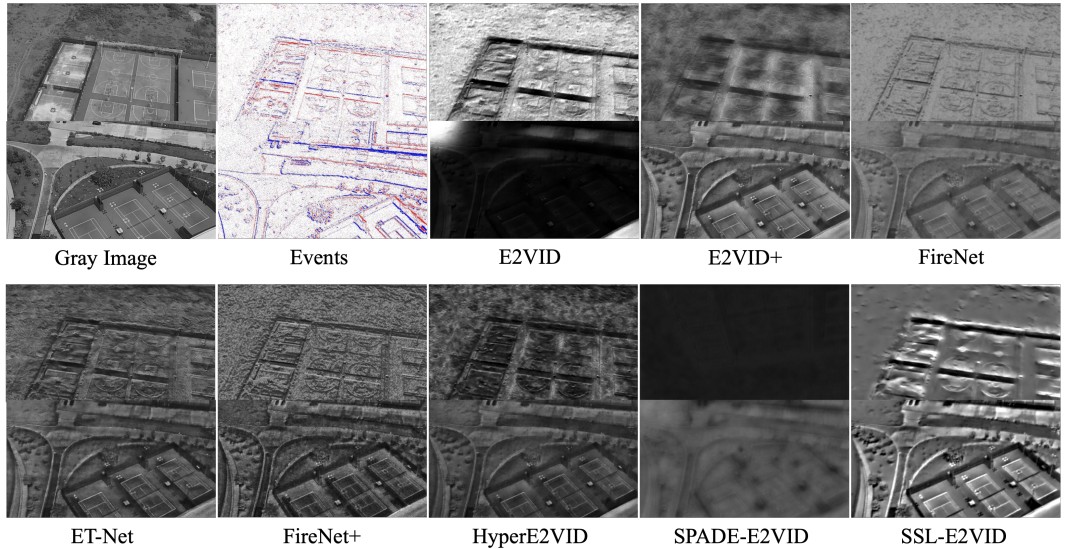

Figure 13: Video reconstruction with E2VID, E2VID+, FireNet, ET-Net, FireNet+, HyperE2VID, SPADE-E2VID, and SSL-E2VID.

## A.7 OTHER TASKS TO EXPLORE

Figure 13 shows the performance of existing event-to-video reconstruction methods E2VID (Rebecq et al., 2019), E2VID+ (Stoffregen et al., 2020), FireNet (Scheerlinck et al., 2020), ET-Net (Weng et al., 2021), FireNet+ (Stoffregen et al., 2020), HyperE2VID (Ercan et al., 2024), SPADE-E2VID (Cadena et al., 2021), and SSL-E2VID (Paredes-Vallés & De Croon, 2021) when applied to our SkyEvents dataset.

