1026
1027
1028
1029
1030
1031
1032
1033
1034
1035
1036
1037
1038
1039
1040
1041
1042
1043

(a) Complex buildings

1044
1045
1046
1047
1048
1049
1050
1051
1052
1053
1054
1055
1056
1057
1058

(b) Low-rise buildings

1059
1060
1061
1062
1063
1064
1065
1066
1067
1068
1069
1070
1071
1072
1073
1074

(c) Open planar scenes

Figure 10: Ground truth for area 1, 2, and 3.

1075
1076
1077
1078
1079

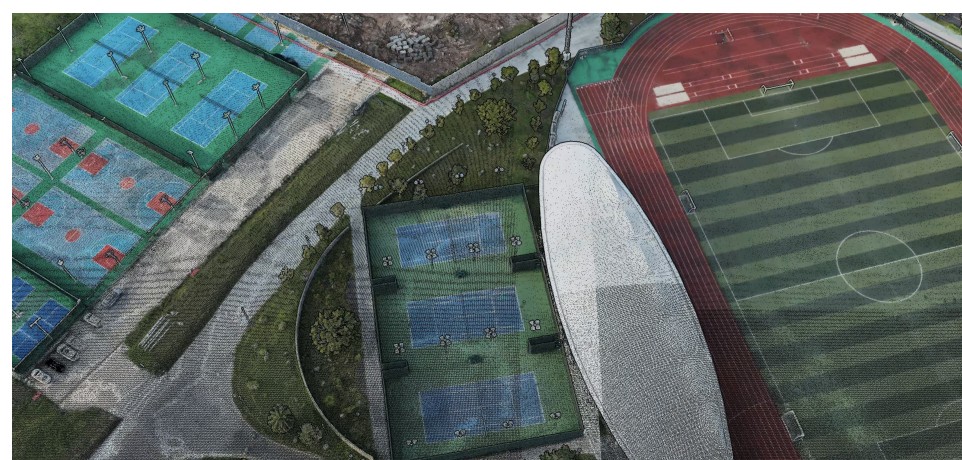

(a) High-rise clusters

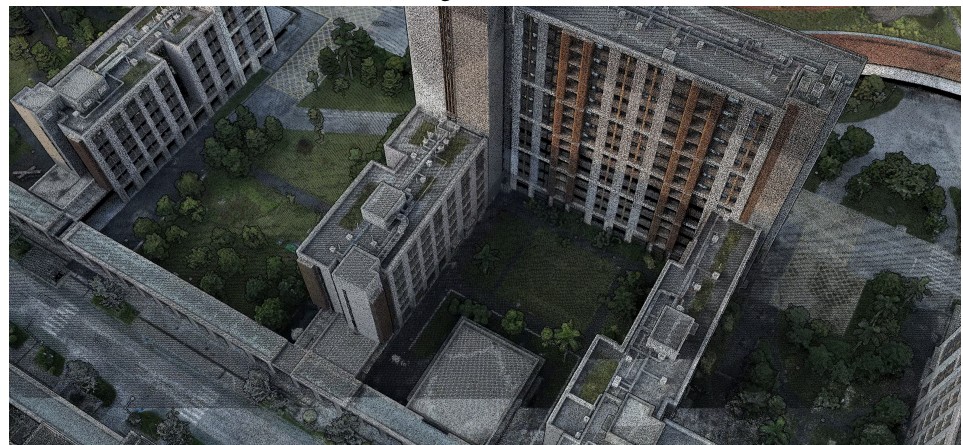

(b) High-rise clusters

Figure 11: Ground truth for area 4,5

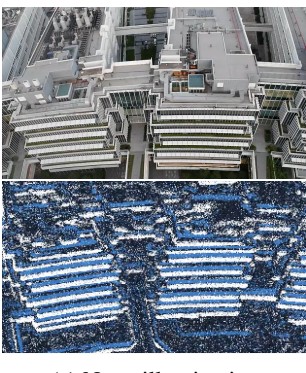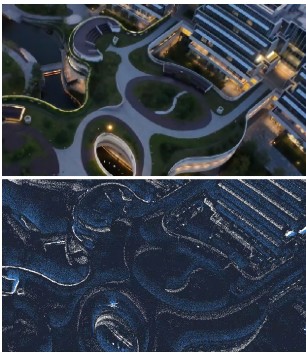

| (a) Noon illumination | (b) Afternoon illumination | (c) Dusk illumination |

Figure 12: Illumination condition samples collected within Area 1 (Main Building complex). Each panel displays synchronized RGB (top) and event camera (bottom) data, demonstrating the variation in visual appearance and event patterns under different lighting conditions while maintaining consistent scene content.

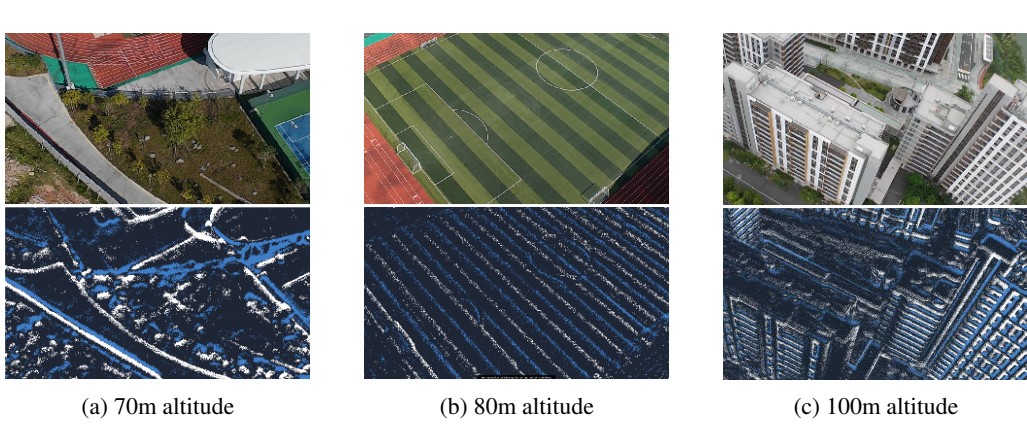

(a) 70m altitude         (b) 80m altitude         (c) 100m altitude

Figure 13: Flight altitude samples collected at the boundary between Area 4 (Playfield) and Area 5 (South Dormitory). Each panel displays synchronized RGB (top) and event camera (bottom) data, showing how spatial coverage and detail resolution vary with altitude while maintaining consistent geographic location.

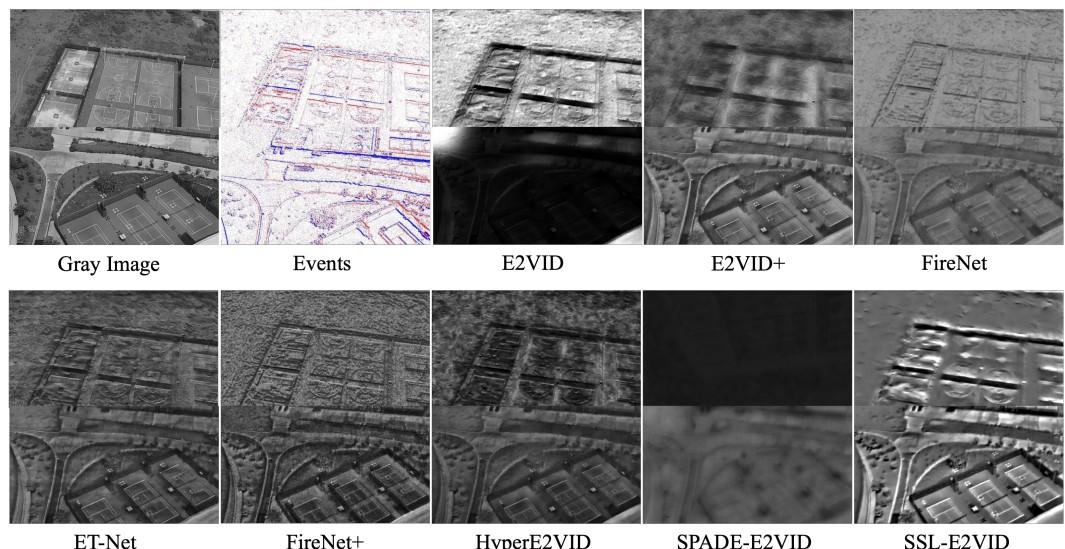

Figure 14: Video reconstruction with E2VID, E2VID+, FireNet, ET-Net, FireNet+, HyperE2VID, SPADE-E2VID, and SSL-E2VID.

## A.5 OTHER TASKS TO EXPLORE

Figure 14 shows the performance of existing event-to-video reconstruction methods E2VID (Rebecq et al., 2019a), E2VID+ (Stoffregen et al., 2020), FireNet (Scheerlinck et al., 2020), ET-Net (Weng et al., 2021), FireNet+ (Stoffregen et al., 2020), HyperE2VID (Ercan et al., 2024), SPADE-E2VID (Cadena et al., 2021), and SSL-E2VID (Paredes-Vallés & De Croon, 2021) when applied to our SkyEvents dataset.