# OpenReview forum: "SkyEvents: A Large-Scale Event-enhanced UAV Dataset for Robust 3D Scene Reconstruction"
_ICLR.cc/2026/Conference — ICLR 2026 Poster_

### Official Review · Reviewer_3NrW · 2025-10-15

**Soundness:** 3
**Presentation:** 2
**Contribution:** 3
**Rating:** 6
**Confidence:** 5

**Summary:**

1. **Originality-wise**: the paper proposes a large-scale UAV event-based dataset for robust 3D reconstruction.
2. **Quality-wise**: the proposed dataset has varied large-scale scenes including different flight path, scenarios types, illumination, and height settings.
3. **Clarity-wise**: the manuscript is clearly written, with well-structured methodology, detailed explanations, and intuitive visualizations that enhance understanding.

**Strengths:**

1. A Pioneering and Highly Valuable Dataset: The most significant contribution of this work is the SkyEvents dataset itself. SkyEvents is not only large in scale (over 8 hours, covering 0.72 km²) but also rich in modalities (RGB, Event, LiDAR) and provides accurate 6-DoF poses, filling a major gap.
2. Solves a Critical Practical Problem: The proposed GTA module directly addresses a very challenging yet common real-world problem: precise timestamp alignment between multiple sensors, especially in low-cost setups without hardware synchronization

**Weaknesses:**

1. **Regarding the GTA Module**: While the paper demonstrates the module's effectiveness indirectly through its positive impact on 3D reconstruction and other downstream tasks, it lacks a more direct and intuitive quantitative evaluation. To compellingly validate the module's performance, I suggest the authors conduct experiments on existing datasets with pre-aligned ground-truth timestamps (e.g., DSEC, MVSEC). This would allow for a direct measurement of the method's alignment error and a quantitative comparison of its effectiveness against other approaches.
2. **Regarding Low-Light Image Generation**: The approach of generating synthetic low-light images is practical for settings where scenes cannot be re-captured. However, aligning event data captured directly under bright sunlight with these synthetically darkened images could introduce a significant data mismatch. Event cameras are known to exhibit different noise characteristics, such as leaky events, in bright conditions compared to true dark environments. This raises concerns about the fidelity of the alignment. I am curious if the authors implemented any specific procedures to address this potential distortion.
3. **Regarding Evaluation Metrics**: I agree that using the same metrics as the original 3DGS is valid for the motion deblur task. However, for low-light or over-exposed conditions, these reference-based metrics may become ineffective due to the quality degradation of the ground-truth images themselves. An evaluation based on a flawed reference is not meaningful. I recommend that the authors incorporate no-reference image quality assessment metrics (e.g., BRISQUE, HyperIQA) to ensure the validity and intuitiveness of the evaluation results.
4. The video visualization is lacking and this reduces the credibility of the real effect.

---
I have listed my concerns, and the score will be adjusted based on the author's response.

**Questions:**

Please refer to Weaknesses part.

---

> ### Author Response · Authors · 2025-12-03
> **Response to Weakness 1**
>
> We sincerely appreciate the reviewer's suggestion to perform a direct evaluation on established benchmarks. Following this recommendation, we conduct additional validation experiments for the GTA module on the MVSEC dataset (which provides ground-truth alignment) as well as our SkyEvents datasets.
>
> The corresponding results are visualized in **Figure 11** of the Supplementary Material.
>
> As shown in **Figure 11**, the event streams aligned by our GTA module exhibit a pixel-perfect correspondence with the RGB frames’ geometric edges.  And the successful alignment on MVSEC confirms that our module generalizes well and achieves high temporal precision.
>
> We believe these qualitative results serve as compelling evidence of the GTA module's effectiveness in correcting timestamp offsets and enforcing geometric consistency.

---

> ### Author Response · Authors · 2025-12-03
> **Response to Weakness 2**
>
> We appreciate the reviewer regarding the potential domain gap caused by event noise characteristics. We address this concern from two aspects: the design of our **loss function** and the nature of the **alignment**.
>
> Our **RER loss** is explicitly designed to be robust to such variations. It does not supervise the network based on individual, sparse events (which are susceptible to specific noise profiles like "leaky events"). Instead, it operates on the accumulated event image, $E(t_1,t_2)$, integrated over a time window, following prior works (e.g., Elite-EvGS).
>
> **Filtering Mechanism**: This accumulation process essentially acts as a spatial-temporal integration, which suppresses high-frequency stochastic noise while reinforcing consistent structural signals.
>
> **Geometric Dominance**: Even though the noise floor differs between bright and low-light conditions, the event density generated by primary geometric edges and texture changes remains significantly higher than the background noise. Consequently, the accumulated image $E(t_1,t_2)$ provides stable geometric constraints that are largely invariant to the specific noise distribution.
>
> Our experimental results in **Table 2** demonstrate significant performance improvements on the synthetic low-light data. This empirically confirms that our GTA module and RER loss effectively bridge the modality gap and remain robust despite the difference in noise characteristics.

---

> ### Author Response · Authors · 2025-12-03
> **Response to Weakness 3**
>
> We thank the reviewer for this thoughtful comment regarding the validity of evaluation metrics in challenging conditions. We would like to address this in two parts: a clarification on our low-light ground truth quality, and the incorporation of no-reference metrics as suggested.
>
> 1.  Clarification on Ground Truth Quality (Synthetic Low-light Setup):  We respectfully wish to clarify the experimental setup detailed in **Section 4.4** and **Section 4.3** "Low-light Image Generation".  To precisely avoid the issue raised by the reviewer—where real-world low-light GT images might suffer from noise or motion blur—we adopted a "Synthetic Low-light" strategy for our quantitative benchmarks:
> Data Construction: We utilize high-quality, well-exposed, and noise-free daytime images as the Ground Truth.
> Input Generation: The corresponding low-light inputs are generated from these high-quality GT images via Gamma correction and linear scaling.
> Under this setup, our reference images are pristine and undegraded. Consequently, full-reference metrics such as PSNR, SSIM, and LPIPS remain mathematically valid and physically meaningful, as they accurately measure the model's ability to recover the original illumination and details from artificially darkened inputs.
>
> 2. Following the reviewer’s valuable recommendation to ensure the intuitiveness of evaluation, we have incorporated No-Reference (NR) Image Quality Assessment metrics—specifically BRISQUE, MANIQA, and NIQE—into our benchmarks.
> We applied these metrics to the Event-to-Video Reconstruction task (a key application of our dataset) in **Figure 7** and **Table 3**. As shown in the updated **Table 3**, the results provide further insight into the performance gap:
> Methods originally trained on ground-level data (e.g., ET-Net, SPADE-E2VID, and SSL-E2VID) exhibit significantly degraded perceptual quality scores when applied to SkyEvents.   This quantitative evidence, supported by the new NR metrics, underscores the significant domain gap and the challenge posed by our aerial, low-light event data, highlighting the necessity for dedicated training on the SkyEvents dataset.
>
> We believe these additional metrics strengthen the comprehensive evaluation of our work.

---

> > ### Author Response · Authors · 2025-12-03
> > **Response to weakness 4**
> >
> > We sincerely thank the reviewer for highlighting this crucial point. We fully agree that regarding large-scale 3D scene reconstruction—particularly for methods based on 3D Gaussian Splatting—static images are insufficient to demonstrate the full potential of the approach. We acknowledge that video visualizations are indispensable for evaluating temporal stability, geometric consistency, and the smoothness of dynamic view synthesis.
> >
> >
> > To address this concern and substantiate the credibility of our experimental results, we have prepared comprehensive video demonstrations. These videos have been uploaded at our project website and will be public after being accepted.

---

### Official Review · Reviewer_r6JB · 2025-10-24

**Soundness:** 2
**Presentation:** 1
**Contribution:** 2
**Rating:** 2
**Confidence:** 4

**Summary:**

This paper introduces SkyEvents, a large-scale 3D scene reconstruction dataset that integrates multi-domain signals, including RGB video, event stream, and LiDAR point clouds. The dataset spans 481 minutes, making it three times longer than the previously largest dataset, EvMAPPER. To handle temporal misalignment between RGB and event signals, this paper proposes a Geometry-constrained Timestamp Alignment (GTA) method. Additionally, it introduces a Region-based Event Rendering (RER) loss to improve event-based rendering optimization. Experiments on SkyEvents include comparisons with and without event inputs in rendering, qualitative analyses of the RER loss, and evaluations against existing reconstruction baselines.

**Strengths:**

- The large-scale real-world SkyEvents dataset is a valuable asset for the scene reconstruction community, providing diverse multi-domain signals.

- Beyond the dataset itself, the methodology for large-scale aerial data collection and the inclusion of hardware details offer practical guidance for future dataset collection efforts.

- The proposed temporal alignment strategy between RGB and event signals enhances the dataset’s usability and convenience for potential users.

**Weaknesses:**

**Incremental Contributions**

a. The proposed Geometry-constrained Timestamp Alignment (GTA) is practically useful for potential users of the SkyEvents dataset. However, in terms of novelty, it appears to be an extension of Dynamic Time Warping (DTW) [1] rather than a fundamentally new formulation. A discussion or comparison with DTW would strengthen the paper.

[1] Berndt, Donald J., and James Clifford. “Using Dynamic Time Warping to Find Patterns in Time Series.” In AAAI Workshop on Knowledge Discovery in Databases, 1994.

b. The motivation and formulation of the Region-based Event Rendering (RER) loss are difficult to follow. Equation (5) is unclear since $\mathcal{C} _ {\theta}$ is not defined anywhere in the paper, and the function $C$ in the definition of $E$ is also unspecified. Even assuming that this loss is intended to minimize the displacement difference of certain features between the initial and final frames (i.e., $\log \mathcal{C} _ {\theta} (\hat{I} _ {t_1}) - \log \mathcal{C} _ {\theta} (\hat{I} _ {t_2})$) and the accumulation of event activities ($E(t _ 1, t _ 2)$), it remains unclear how this formulation guides or benefits event rendering optimization.

c. Excluding GTA and RER, the main contribution is limited to the dataset itself. However, the supplementary information supporting the usefulness of SkyEvents is quite limited. The paper should elaborate on the design philosophy behind the dataset—specifically, why certain characteristics were selected and how they differ from existing datasets. Furthermore, a comprehensive comparison with other datasets in terms of these characteristics is necessary to better demonstrate the dataset’s uniqueness and value.

**Immature Presentation**

a. As mentioned earlier, some functions used in the definition of the RER loss (i.e., $C_{\theta}$ and $C$) are not defined. Furthermore, the connection between the mathematical definition of RER and its underlying design intuition is not sufficiently explained, making it difficult to understand the motivation behind this formulation.

b. The related work section primarily focuses on datasets. If GTA and RER loss are intended to be core contributions of the paper, a more comprehensive review of related methods should be included to demonstrate the distinct features of the proposed methods compared to existing ones, and to clarify why these components are crucial for the given task, i.e., multi-modal dataset construction and scene reconstruction.

c. The evaluation structure requires revision.

- It is unclear what the main results of the paper are in Section 4.4 (Results), and how these differ from the results presented in Section 4.5.
The authors should clarify what readers are expected to focus on in the baseline comparison shown in Figure 6.

- In Figure 4, it is not explained why comparisons are made with and without GS, and what specific conclusion can be drawn from this comparison.

**No Benefit in Evaluation**

a. In Figure 4, it is unclear what benefit the RER loss provides; the difference between the results with and without RER is not evident.

b. In Table 2, the use of event data does not consistently improve performance. The purpose or takeaway of this comparison should be clarified—what should the reader conclude about the effectiveness of incorporating event signals?

**Questions:**

Is it typical to use LiDAR point clouds as ground truth for scene reconstruction? While LiDAR provides higher accuracy than other sensing modalities, it still contains measurement noise and unobserved regions from the camera’s viewpoint, which may lead to inaccurate quantitative evaluation. How is this limitation addressed or considered in the evaluation process?

**Details Of Ethics Concerns:**

I don't find any ethics concern.

---

> ### Author Response · Authors · 2025-12-03
> **Response to Weakness (Incremental Contributions)**
>
> ## Response to Weakness a
>
> We thank the reviewer for this insightful comment. We agree that GTA is conceptually related to time-series alignment methods such as Dynamic Time Warping (DTW) [1]. Our intention is not to present GTA as a fundamentally new time-series paradigm, but rather **as a practical, geometry-constrained alignment module tailored to the SkyEvents multi-sensor UAV platform.**
>
> GTA is designed for a specific and challenging setting: UAV flights at 50–100 m altitude, under low light / motion blur, with no access to large calibration targets such as checkerboards, while requiring consistent alignment of RGB, event, and LiDAR in a shared coordinate frame and time axis. **In this context, classical DTW and related methods would need substantial adaptation, and they typically ignore the camera geometry that is crucial in our setting.**
>
> Concretely, GTA differs from classical DTW in several key aspects:
>
> 1. Objective and model of time warping.
> DTW usually assumes two unconstrained discrete time series and searches for an arbitrary monotonic warping path on a 2D time grid, purely in the time-series domain. In contrast, GTA does not search for an arbitrary warping path. **We exploit the fact that our acquisition system already provides hardware-level coarse synchronization, and we restrict the problem to a parametric, weak time warp (e.g., a global offset or small drift) within a narrow candidate range.** The goal is to refine a nearly synchronized system, not to freely warp two unrelated sequences.
>
> 2. Geometry-driven alignment cost instead of signal-domain distance.
> DTW typically minimizes a distance between signal values (e.g., intensity, feature distance) at matched time indices. GTA, however, uses an explicit multi-view geometric objective. We first perform dense matching between event representations and RGB images and then select high-confidence matches with reprojection error below a small threshold (e.g., 0.01). The time offset and extrinsic parameters are optimized to minimize geometric reprojection error under the camera projection model, rather than a purely signal-domain alignment loss. In other words, **temporal alignment in GTA is solved through the lens of 3D geometry and camera calibration, which is particularly important in noisy, low-light event streams.**
>
> 3. Coupled spatial/temporal calibration.
> DTW addresses only temporal alignment. GTA is explicitly formulated to jointly estimate RGB–event extrinsics and residual time offset under UAV flight conditions, and then propagate this alignment to LiDAR via LiDAR-centered SfM. Thus, **it is not a generic time-series algorithm, but a multi-sensor calibration and synchronization module that embeds time alignment into a geometric calibration pipeline.**
>
> Given these points, we believe a more accurate way to position GTA is as a geometry-constrained, system-specific specialization inspired by time-series alignment ideas.
>
> ## Response to Weakness b
>
> We sincerely thank the reviewer for pointing out the lack of clarity regarding the Region-based Event Rendering (RER) loss. We apologize for the undefined notations in the original manuscript.
>
>
>
> To address your concerns, we have made the following revisions:
>
>
>
> 1. We have explicitly defined all symbols in **Equation (5)**, including the previously undefined variable and the specific function used. The mathematical formulation is now rigorous and self-contained.
>
> 2. We have rewritten the motivation sentences to better explain how the RER loss works. The core idea is to utilize the high temporal resolution of events to constrain the intensity changes in the rendered RGB images.
>
> 3.  We clarified that using event data to supervise novel view synthesis is a standard and effective practice in recent Event-based Gaussian Splatting research. By enforcing consistency between the accumulated events and the rendered brightness changes, the model can achieve better geometric accuracy. **We have added relevant citations (Elite-EvGS) to support this formulation.**

---

> > ### Author Response · Authors · 2025-12-03
> > **Response to Weakness (Incremental Contributions)**
> >
> > **Response to Weakness c**
> >
> > We thank the reviewer for this constructive comment. We agree that, beyond basic statistics and the fact that SkyEvents is (to the best of our knowledge) the first large-scale multi-view event dataset explicitly designed for 3D reconstruction, the paper should better articulate the dataset’s design philosophy and clearly position it against existing datasets.
> >
> > In the current version, we report scene counts, numbers of RGB/event frames, resolutions, etc., but we agree this is not sufficient to convey why SkyEvents is structured the way it is and what long-term research problems it is intended to support. In the revision, we have therefore expanded both the textual description and the comparison in Table 1.
> >
> > First, we now more systematically explain the design and processing choices behind SkyEvents. The dataset is not constructed by simply mounting an event camera onto a multi-view platform; instead, the acquisition conditions and annotation pipeline are deliberately tailored around the strengths and challenges of event data in real 3D reconstruction scenarios. For example, we deliberately select scenes with different spatial extents and flight patterns (varying altitudes, baselines, and trajectories) to cover regimes where traditional RGB-based 3D reconstruction and Gaussian Splatting pipelines tend to degrade, such as low-light, high-speed motion, and large parallax. We adopt a multi-view capture strategy with overlapping fields of view so that the events are naturally usable not only for deblurring or low-light enhancement, but also for multi-view geometry and full 3D reconstruction. Through joint acquisition and calibration with LiDAR, we place RGB, events, and high-precision geometry in a unified coordinate frame and time axis, enabling a family of geometry-centric tasks, including monocular/event-based depth estimation, 3D reconstruction, and inverse rendering.
> >
> > Second, we fully agree with the reviewer that these characteristics must be contrasted more explicitly with existing datasets. In the revised manuscript, we have enriched Table 1 and its surrounding discussion to provide a more comprehensive comparison along dimensions such as: whether a dataset supports globally consistent multi-view geometry, whether it provides LiDAR or dense depth supervision, whether RGB and event sensors have aligned fields of view, whether the data is captured under challenging conditions relevant to 3D reconstruction (e.g., low light, strong motion blur), and whether the dataset is explicitly designed for 3D tasks rather than mainly 2D or SLAM-oriented benchmarks. Within this comparison, we highlight that SkyEvents is designed from the outset as a 3D-focused, UAV-based event benchmark with synchronized RGB–event–LiDAR, drift-free global poses, and LiDAR-derived dense depth, which differentiates it from prior event datasets that typically lack either multi-view structure, high-precision geometry, or UAV-scale outdoor capture tailored to 3D reconstruction.
> >
> > Finally, we also agree with the reviewer’s perspective on contributions: even if one ignores GTA and RER, the primary value of the paper should lie in the dataset’s structure, coverage, and the range of supported tasks. To reflect this, we have adjusted the narrative to more clearly position GTA and RER as supporting tools and reference baselines built on top of SkyEvents, rather than as the main innovations. The main contribution is now framed around the dataset design, its multi-modal, geometry-aware acquisition, and the new 3D event-based evaluation scenarios it enables.
> >
> > We hope these clarifications and additions make the uniqueness and usefulness of SkyEvents more concrete and convincing.

---

> ### Author Response · Authors · 2025-12-03
> **Response to Weakness (Immature Presentation)**
>
> **Response to Weakness a**
>
> Thanks, we have revised it in the updated manuscript. Please refer to "Response to Weakness b" in "Incremental Contributions"
>
> **Response to Weakness b**
>
> We appreciate the reviewer’s comments and would like to clarify that the core contribution of this paper is the SkyEvents dataset itself, rather than generic alignment algorithms or rendering losses.
>
> SkyEvents is designed to fill a gap in the community: the lack of a large-scale, multi-modal UAV event dataset with accurate 3D geometry for robust 3D scene reconstruction. Our primary goal is to provide high-quality data infrastructure; GTA and RER are introduced as supporting components that make the dataset practically usable and evaluable, not as standalone state-of-the-art methods.
>
> (i). GTA is a geometry-constrained timestamp alignment module that solves a practical calibration/synchronization problem specific to our UAV multi-sensor platform (RGB + event + LiDAR). It is a prerequisite tool for constructing a high-quality multi-modal dataset, not a general-purpose temporal alignment method we claim as a main contribution.
>
> (ii). RER loss is introduced as a baseline component to demonstrate how SkyEvents can be used within a 3D Gaussian Splatting framework; it serves to establish a reference benchmark on our dataset rather than to compete with existing rendering or loss formulations in general.
>
> Therefore, it is intentional that the related work section is primarily focused on datasets, consistent with the positioning of SkyEvents as a dataset paper.
>
> In the revised manuscript, we have (i) clarified in the Introduction and contribution summary that the dataset is the central contribution and GTA/RER are auxiliary modules built to support multi-modal dataset construction and benchmarking, and (ii) adjusted the method sections to de-emphasize algorithmic novelty and to present GTA and RER explicitly as tools that enable and showcase the dataset, rather than as core methodological contributions.
>
> **Response to Weakness c**
>
> **1. Roles of Sections 4.4 and 4.5.**
>
> In the revised manuscript, we explicitly clarify that these two sections play different but complementary roles in evaluating the usefulness of SkyEvents:
>
> (i) **Section 4.4** (Results on 3D Reconstruction) focuses on the core task of the dataset: 3D scene reconstruction. Here, the goal is to demonstrate that SkyEvents, together with our alignment module (GTA) and event-aware loss (RER), supports high-quality 3D Gaussian Splatting–based reconstruction. This section should be read as the main result showing the usability and effectiveness of SkyEvents for its primary 3D reconstruction use case.
>
> (ii). **Section 4.5** (Other Tasks to Explore) is intended to highlight the versatility and difficulty of the dataset beyond the core 3D task. In this section, we evaluate representative methods on monocular depth estimation and event-to-video reconstruction. These experiments are not meant to claim that our methods outperform existing SOTA approaches; instead, they are meant to probe how current SOTA models generalize to SkyEvents and to show that our dataset poses non-trivial challenges for existing algorithms.
>
> **2. Results on in Figure 6 (depth baselines).**
>
> **Figure 6** presents baseline results for depth estimation using existing state-of-the-art methods on SkyEvents. The key takeaway we intend for readers is not that our own method dominates these baselines, but rather:
>
> (i) Existing SOTA depth models, when applied to our UAV event/RGB setting, exhibit visible artifacts and loss of fine structures, especially in complex aerial scenes.
>
> (ii) This behavior indicates that current methods have been trained and tuned mostly on simpler or non-UAV event datasets and do not generalize well to the large-scale, multi-view, outdoor scenarios captured in SkyEvents.
>
> In other words, **Figure 6** is meant to illustrate that SkyEvents exposes clear failure modes and limitations of existing methods, thereby underscoring both the scarcity of suitable training data for UAV event-based 3D perception and the research value of our dataset as a challenging benchmark.
>
> **3. Revisions made**
>
> Following the reviewer’s suggestion, we have revised the text around **Sections 4.4 and 4.5** to:
>
> (i). Explicitly state that **Section 4.4** contains the main reconstruction results, while **Section 4.5** provides exploratory, task-oriented evaluations to showcase versatility and difficulty.
>
> (ii). Clarify the motivation for including monocular depth estimation and event-to-video reconstruction.
>
> (iii). Add guiding text around **Figure 6** to explicitly highlight what readers should look for (the failure patterns and missing fine details in existing methods), and to explain how these observations support our claim that SkyEvents is a challenging and valuable benchmark.

---

> > ### Author Response · Authors · 2025-12-03
> > **Response to Weakness (Immature Presentation)**
> >
> > **Response to Weakness d**
> >
> > We thank the reviewer for this valuable feedback. We realize that the labeling and interpretation of Figure 4 are not sufficiently clear in the original manuscript. We have addressed this as follows:
> >
> > **1. Clarification of Experimental Setup**
> >
> > The comparison in **Figure 4** is conducted between the Baseline (RGB-only 3DGS) and Ours (RGB + Event-enhanced 3DGS with RER loss).
> >
> > Purpose: This comparison aims to visually validate the efficacy of incorporating the Event Modality and our RER Loss in addressing motion blur.
> >
> > Context: Low-altitude UAV flights often involve rapid motion, resulting in blurred RGB images. Standard 3DGS methods, which rely solely on RGB inputs, inevitably produce blurred geometry and texture under these conditions. We aim to demonstrate that the event stream from the SkyEvents dataset serves as a critical complementary information source to correct these artifacts.
> >
> >
> > **2. Specific Conclusion**
> >
> > The core conclusion drawn from this experiment is that the introduction of the RER loss significantly enhances reconstruction sharpness and detail fidelity.
> >
> > As shown in the updated **Figure 4**, the Baseline method inherits the blur from the input RGB images, resulting in fuzzy edges.
> >
> > In contrast, our method leverages the high temporal resolution of event data to recover lost high-frequency details (e.g., sharp building edges and textures), proving the capability of event data to enhance 3D reconstruction under extreme conditions.
> >
> > **3. Revisions**
> >
> > Following your suggestion, we have made the following changes:
> >
> > Updated **Figure 4**: We have refined the visualization and labels to clearly distinguish between the baseline and our method, explicitly highlighting the deblurring effect.
> >
> > Enhanced **Section 4.4**: We have expanded the text in **Section 4.4** to include a detailed qualitative analysis of these results, explicitly discussing the contribution of event data in mitigating blur artifacts.

---

> > > ### Author Response · Authors · 2025-12-03
> > > **Response to Weakness (No Benefit in Evaluation)**
> > >
> > > **Response to Weakness a**
> > >
> > > We apologize for the lack of clarity in the original visualization and thank the reviewer for pointing this out. We have updated **Figure 4** in the revised manuscript to better highlight the contribution of the RER loss.
> > >
> > > As illustrated in the revised figure, the qualitative difference is now more evident:
> > >
> > > **Smoothness:** With the integration of RER loss, the reconstructed novel views exhibit significantly smoother surfaces in homogeneous regions, effectively suppressing high-frequency noise and artifacts.
> > >
> > > **Detail Preservation:** Crucially, this smoothness does not come at the cost of blurring; the model continues to preserve sharp geometric details and edges.
> > >
> > > This comparison confirms that the RER loss effectively regularizes the 3D structure, balancing surface smoothness with detail fidelity.
> > >
> > > **Response to Weakness b**
> > >
> > > Thank you for pointing this out. We agree that the message of **Table 2** needs to be stated more explicitly, both in the text and in the caption.
> > >
> > > From a quantitative perspective, the pattern in **Table 2** is more consistent than it may appear at first glance. Across all six scene and condition combinations, adding events always increases PSNR, with gains between about 0.06 and 0.62 decibels. LPIPS decreases in five out of six rows, and in the remaining case it increases by only about 0.006, which is a very small change at this scale. SSIM remains broadly comparable, with variations on the order of a few thousandths. In other words, events never cause a collapse in any metric, and they bring clear PSNR and LPIPS gains precisely in the regimes where motion blur and exposure changes are severe, especially for the blurred setting and for the larger Scene 2.
> > >
> > > At the same time, **Table 2** is not intended to claim that every single scalar metric will strictly improve in every configuration. There is a trade off between sharper reconstruction and global smoothness that metrics such as SSIM and LPIPS capture only imperfectly. In the small Scene 1 with the additional gamma recovery kernel, for example, event supervision sharpens fine edges and structures, which improves PSNR but slightly changes the local contrast pattern, leading to a marginal LPIPS increase. As **Figure 4** illustrates, these cases still show visibly sharper facades, fewer double contours, and reduced ghosting when events are used, even when aggregate metrics move only slightly.
> > >
> > > We have revised the text around **Table 2** to make this takeaway explicit. The purpose of the comparison is to show that event cues act as a complementary high frequency constraint for 3D Gaussian splatting: they stably improve deblurring and low light reconstruction on challenging and large scale scenes, while keeping standard image quality metrics at least on par, and often better, than RGB only baselines. We will also add a short sentence that directs readers from **Table 2** to the corresponding visual comparisons, so that the quantitative and qualitative evidence are read together rather than in isolation.

---

> > > > ### Author Response · Authors · 2025-12-03
> > > > **Response to Question**
> > > >
> > > > We thank the reviewer for raising this important point. Using LiDAR as a geometric reference for depth and reconstruction evaluation is a **standard practice** in 3D vision and autonomous driving benchmarks, because survey-grade LiDAR typically offers significantly higher geometric accuracy than image-based methods and provides a stable, metrically calibrated coordinate frame for comparing different algorithms.
> > > >
> > > > In SkyEvents, we follow this common practice but rely on a survey-grade LiDAR system and a dedicated acquisition scheme:
> > > >
> > > > (i). Our LiDAR system is specified to achieve **<2 cm measurement error at 150 m range**, which is substantially smaller than the reconstruction errors we typically observe from RGB/event-based methods in the same scenes.
> > > >
> > > > (ii). LiDAR acquisition is performed in **separate, dedicated flights** with denser flight lines than the RGB/event captures. We design the LiDAR trajectories to ensure a point density of **at least ~200 points/m²** over the area of interest, which greatly reduces gaps and coverage issues and provides a dense, high-quality reference surface over the whole mapped region.
> > > >
> > > > In the evaluation, we treat LiDAR not as an absolutely perfect ground truth, but as a **high-accuracy reference geometry**:
> > > >
> > > > (i). Depth maps are generated by projecting the LiDAR point cloud into the camera views; evaluation is restricted to pixels where LiDAR has valid, reliable returns. Regions that are unobserved or poorly sampled by LiDAR are excluded from the metrics, so they do not bias the quantitative evaluation.
> > > >
> > > > (ii). Given the centimeter-level LiDAR noise and high point density, the residual LiDAR error is significantly smaller than the differences between reconstruction methods we compare, making it suitable as a reference for relative performance evaluation.
> > > >
> > > > We will clarify these points in the revised manuscript, explicitly stating that (i) we follow the established practice of using survey-grade LiDAR as reference geometry, and (ii) we mitigate its limitations through dense acquisition and masking to LiDAR-observed regions in the evaluation protocol.

---

### Official Review · Reviewer_WeKv · 2025-10-30

**Soundness:** 2
**Presentation:** 2
**Contribution:** 1
**Rating:** 2
**Confidence:** 4

**Summary:**

This paper introduces SkyEvents, a large-scale multimodal UAV dataset incorporating RGB, event, and LiDAR data, with particular emphasis on the role of event cameras for 3D scene reconstruction in urban environments. While the dataset is valuable in its focus on event-based sensing under challenging conditions, the paper's primary contribution remains at the data collection and preliminary fusion level.

**Strengths:**

The strengths of this work include the creation of the comprehensive SkyEvents dataset, which features diverse challenging conditions and multimodal data (RGB, event, LiDAR). It potentially enables improved 3D reconstruction and perception algorithms, introduces effective synchronization (GTA) and rendering techniques (RER), and demonstrates the beneficial impact of event data in various challenging scenarios like low-light and motion blur.

**Weaknesses:**

The main limitation is the lack of in-depth task-specific analysis demonstrating the concrete benefits of event data for downstream perception tasks such as depth estimation, semantic segmentation, or real-time scene understanding. The experimental results are mostly qualitative and do not sufficiently quantify the actual improvements brought by event modalities. As a result, the paper falls short of establishing clear performance gains or providing comprehensive evaluations that would justify its impact as a benchmark resource.

Furthermore, insufficient details on calibration, synchronization, and data processing hinder reproducibility and broader adoption. Without extensive task-oriented experiments and rigorous benchmarking, the work remains at an early exploratory stage, limiting its immediate utility for advancing event-based perception research.

Overall, while the dataset is a promising resource, the paper does not demonstrate enough analysis or experimental validation to warrant publication in its current form. Significant additional work is necessary to substantiate the practical advantages of the provided data for perception tasks.

**Questions:**

1. How did you address synchronization challenges between RGB, event, and LiDAR sensors?

---

> ### Author Response · Authors · 2025-12-03
> **Resonse to Weakness 1：specific analysis**
>
> We appreciate the reviewer’s concern about task specific analysis and agree that a benchmark paper should clearly quantify the benefits of events rather than rely only on visual comparisons.
>
> In the current version, we deliberately focused on geometry oriented perception tasks, in particular depth estimation and three dimensional reconstruction, because our survey grade LiDAR allows us to construct accurate depth ground truth. Based on these LiDAR derived depth maps, we already evaluate depth estimation models that use RGB only and models that additionally consume events, and we report quantitative improvements brought by the event modality in terms of depth error metrics. We acknowledge that this part was not sufficiently emphasized in the original submission, and we have clarified and expanded the corresponding analysis in the revision, with explicit references to the relevant **Section 3.4**, **Section 4.4**, **Section 4.5**, **Figure 2**, **Figure 6**, **Table  1**, and **Table  3**.
>
> Beyond these, we also study the role of events inside non event native three dimensional Gaussian splatting pipelines. Through the GTA alignment module and the RAR module with the proposed RER loss, we systematically compare low light and motion blur scenarios with and without event input, under identical training and evaluation protocols. **The revised manuscript highlights that the inclusion of events yields consistent quantitative gains in reconstruction quality, geometric stability and rendering sharpness, as measured by standard image and geometry metrics, which provides concrete evidence that events help mitigate the degradation of conventional 3D Gaussian splatting under extreme conditions.**
>
> In response to the specific comment about task specific analysis, we have added an additional **event driven depth estimation experiment** in the new version (see **Section 4.5** and **Figure 6**). Using the unified LiDAR depth ground truth and report their differences on depth error measures. For higher level downstream tasks such as semantic segmentation or real time scene understanding, we fully agree that events are promising. **However, these tasks require large scale semantic annotations and task specific network designs, which would substantially broaden the scope of this first dataset paper.** We will put these taskes as important next steps that can be built on top of the dataset and the geometric benchmarks established here.

---

> ### Author Response · Authors · 2025-12-03
> **Resonse to Weakness 2：calibration and synchronization**
>
> We appreciate the reviewer’s concern about reproducibility and practical utility. We fully agree that clear calibration, synchronization, and processing details are essential for a dataset to be broadly adopted. In the revision we have therefore expanded **Section 3** to give a more explicit description of our data curation pipeline, with particular emphasis on the **GTA** module, **temporal synchronization**, and **LiDAR alignment**.
>
> For event and RGB calibration, we explain **why standard checkerboard based calibration is not feasible in our setting**. At typical UAV flight altitudes from **50 to 80** meters, even a carefully prepared checkerboard only helps to refine intrinsics in controlled conditions, while in real flights the event data are extremely sparse and blurry and it is practically impossible to deploy a checkerboard at the working distance in a stable way. **Instead, the GTA module follows the same principle as checkerboard calibration but uses real scene features.** For synchronized pairs of event images and RGB frames, we perform **dense feature matching** to obtain about twenty thousand candidate correspondences, then filter them by reprojection error to keep roughly one thousand high confidence matches, and estimate the relative pose and extrinsics from these points.
>
> For temporal synchronization, we clarify that the **acquisition platform includes a small trigger script that fires both the event and RGB cameras while recording a shared UTC time stamp at millisecond resolution**. **These hardware time stamps provide a coarse alignment, which is then refined by the GTA module using consistency between event streams and RGB intensity changes.** In practice this reduces residual temporal misalignment to within approximately three milliseconds. For RGB and LiDAR, we add a detailed description of the joint SfM and extrinsic calibration procedure that brings all sensors into a common global frame, and of how we project the aligned LiDAR into image space to obtain pixel level depth ground truth.

---

> > ### Author Response · Authors · 2025-12-03
> > **Resonse to Question 1：synchronization challenges between RGB, event, and LiDAR sensors**
> >
> > We thank the reviewer for raising this important point. We handle RGB–event and LiDAR–camera alignment in two stages: (i) spatiotemporal calibration between RGB and events, and (ii) alignment to the LiDAR coordinate frame.
> >
> > **For RGB and event cameras, we combine hardware-level coarse synchronization with a geometry-constrained refinement:**
> >
> > 1. At acquisition time, both sensors are triggered under a shared time base, providing coarse temporal alignment via common timestamps.
> >
> > 2. On top of this, our GTA module performs joint spatial calibration and fine time alignment directly in real outdoor scenes.
> >
> > 3. We then filter correspondences by reprojection error, retaining only high-confidence matches with very low geometric error, and jointly solve for the RGB–event extrinsics and the residual time offset.
> >
> > **For LiDAR–RGB alignment, we adopt a LiDAR-centric reconstruction strategy and then propagate the alignment to events:**
> >
> > 1. We first take the LiDAR poses and trajectory as the reference and, in the LiDAR coordinate system, render RGB views at the LiDAR poses. These LiDAR-rendered RGB images are treated as anchor views whose poses are fixed.
> >
> > 2. Next, we perform a joint aerial triangulation / SfM optimization over the real RGB images and the LiDAR-rendered RGB views. During bundle adjustment, the LiDAR anchor poses remain fixed, and we solve for all real RGB camera poses directly in the LiDAR global coordinate frame.
> >
> > 3. Since the event camera has already been geometrically calibrated and temporally aligned to the RGB camera via GTA, its poses can be obtained by applying the known rigid transform from RGB to event. As a result, RGB, event, and LiDAR all share a unified 3D coordinate system and a consistent time reference.

---

### Official Review · Reviewer_jye9 · 2025-10-30

**Soundness:** 3
**Presentation:** 3
**Contribution:** 3
**Rating:** 6
**Confidence:** 5

**Summary:**

This work proposes a large-scale multimodal UAV dataset (i.e., SkyEvents) for 3D scene reconstruction, which includes RGB images, events, and LiDAR data. This newly built dataset provides 45 hybrid sequences considering light changes, scene diversity, and flight altitudes. Besides, the authors present a GTA module to align timestamps between two streams and design an RER loss function for supervising the rendering optimization. I believe the large-scale dataset will provide a challenging benchmark for event-based 3D reconstruction using agile drones.

**Strengths:**

1. The topic of event-based 3D reconstruction using agile drones is very interesting.

2. The large-scale multimodal UAV dataset, which includes events, frames, and point clouds, will benefit the event-based vision community.

3. The writing is clear and easy to understand.

**Weaknesses:**

1. Figure 1 in the manuscript is very impressive and nicely illustrates large-scale 3D scene reconstruction. The authors are encouraged to include more examples of large-scale 3D scenes in the visualization results, ideally covering high-speed or low-light scenarios, to highlight the advantages of event-based modalities compared to RGB frames and RGB–event multimodal settings.

2. As a dataset or benchmark paper, it would be beneficial to evaluate more existing event-based 3D reconstruction methods for comparison. At least incorporating some open-source algorithms would make the dataset more useful and accessible to a broader research community.

3. Given that this work mainly focuses on presenting a large-scale dataset, it might not be ideal to introduce new components such as the GTA module and RER loss in the same paper, as this could make the focus of the paper less clear.

4. The inclusion of LiDAR point clouds in the dataset is an excellent idea. However, the manuscript currently lacks sufficient experimental results or demonstrations showing the value or impact of the point cloud data.

5. For a dataset paper, Table 1 is particularly important. The authors are encouraged to further enrich this table and clearly articulate the unique contributions and advantages of their dataset compared to existing ones. This would make the paper more solid and convincing.

6. Minor suggestions in the future work:

a. In the Related Work section, the titles of Sections 2.1 and 2.2 could be revised, as their contents overlap. Section 2.2 could be renamed to Event-based 3D Reconstruction for clarity.

b. For large-scale UAV datasets, it is suggested that future work could employ stereo event cameras, as in [1,2]. The authors might even consider multi-UAV collaboration to collect coordinated datasets for 3D scene reconstruction, potentially leveraging event camera simulators such as [3].

c. The authors could also refer to and cite more recent works on event-based 3D reconstruction, e.g., [4].

[1] Active Event-based Stereo Vision, CVPR 2025.

[2] Enhanced event-based dense stereo via cross-sensor knowledge distillation, ICCV 2025.

[3] Physical-based event camera simulator, ECCV 2024.

[4] EvaGaussians: Event stream assisted Gaussian splatting from blurry images, ICCV 2025.

**Questions:**

1. How were the event camera (e.g., EKV4) and the frame-based camera (e.g., DJI Action 4) physically integrated and synchronized to ensure spatiotemporal alignment, particularly under the vibrations experienced during UAV flight?

2. The authors suggest that the primary motivation for introducing an event camera in UAV scenarios is to address high-speed motion blur. Under what specific conditions of flight altitude and velocity would an event camera provide a clear advantage over a standard 20 FPS RGB camera, which is susceptible to such motion blur?

---

> ### Author Response · Authors · 2025-12-03
> **Resonse to Weakness 1**
>
> Thank you for the thoughtful suggestion. Our dataset is indeed designed to stress event-based sensing in large-scale UAV scenarios, and we already include experiments that go beyond static visualization. In addition, we evaluate event-based monocular depth estimation from UAV viewpoints in **Figure 6**, and the observed failure cases of existing methods highlight the potential of our dataset as a benchmark for single-view event-based 3D vision.
>
> Regarding visual results, **Figure 4** specifically targets severe motion blur in RGB-only inputs leads to loss of geometric edges and texture details, whereas incorporating events (Improved-GS + RER) enables the model to recover these missing structures. **Figure 5** focuses on low-light conditions: leveraging the high dynamic range of events, our RGB–event model substantially reduces rendering artifacts and improves image sharpness compared to the SOTA Luminance-GS baseline.
>
> To further strengthen this aspect and directly respond to the reviewer’s comment, we added an additional set of large-scale visualizations in the revised version.

---

> > ### Author Response · Authors · 2025-12-03
> > **Resonse to Weakness 2**
> >
> > We appreciate this suggestion and agree that a dataset and benchmark paper benefits from evaluating a diverse set of existing methods, especially those that are open source. In designing our experimental protocol, we started from 3D Gaussian splatting methods that already perform strongly in urban rendering. In particular, we adopt Luminance-GS (Cui et al., 2025) and Improved-GS (Deng et al., 2025) as our base models, and introduce event information through the proposed Region wise Event Rendering loss.
> >
> > As reported in **Table 2** and illustrated in **Figure 4** and **Figure 5**, we compare these SOTA baselines in two settings, with and without event input. The results show that adding events significantly improves robustness under low illumination and motion blur, while keeping the underlying 3DGS backbone unchanged. This provides future researchers with a practical and high performance baselines that can be directly extended on top of SkyEvents.
> >
> > Follow your suggestion, we additionally evaluate a broader set of event based vision tasks in **Section 4.5**. For event to video reconstruction, we benchmark eight open source algorithms, including E2VID, FireNet, SSL E2VID, and HyperE2VID, as summarized in **Table 3** and **Figure 7**. Their performance on UAV aerial data indicates that there is still substantial room for improvement in this setting. For monocular depth estimation, we evaluate the **E2Depth** model on our sequences and show in **Figure 6** that SkyEvents poses clear geometric challenges for current methods.
> >
> > During rebuttal period, we also surveyed and attempted to adapt several existing event based 3D reconstruction pipelines. Many of these systems, however, rely on their own tightly integrated data formats and event representations, for example **pre rendered event images**, .**npy tensors**, or **specific voxel grids**. Achieving a fair and stable comparison on our raw event streams and unified calibration would require rewriting data loaders, aligning preprocessing, and carrying out extensive tuning and validation. Under the current revision schedule, we could not complete this work to a standard that we consider reliable, and we prefer not to report baselines that might be unfair.
> >
> > In parallel with the public release of SkyEvents, we plan to provide standardized event data loaders and format conversion tools that make it easier to plug in representative open source event based 3D reconstruction methods.

---

> > > ### Author Response · Authors · 2025-12-03
> > > **Resonse to Weakness 3**
> > >
> > > Thank you for your constructive concern regarding the introduction of new components such as the GTA module and RER loss. We would like to clarify that these components are not introduced as standalone algorithmic contributions but as necessary tools for this dataset. The GTA module ensures data quality by addressing the inherent issues in high-speed UAV flight and misaligned sensor synchronization. The RER loss, on the other hand, serves as a baseline to validate the dataset's effectiveness in large-scale 3D reconstruction, particularly in challenging scenarios like low-light and motion blur. These tools are part of the dataset's curation pipeline, not independent algorithms. We have revised the manuscript to emphasize these components as supportive tools for the dataset and benchmarks, rather than focusing on them as novel contributions. This will ensure that the core focus of the paper remains on the dataset itself and its potential applications. We also adjust the introduction and method sections to highlight these clarifications and focus the narrative on the dataset’s design, annotation processes, and evaluation protocols.

---

> > > > ### Author Response · Authors · 2025-12-03
> > > > **Resonse to Weakness 4**
> > > >
> > > > We thank the reviewer for highlighting this point. We would like to clarify that in SkyEvents, LiDAR is not merely an optional extra modality, but a core geometric backbone that enables high-precision supervision and drift-free poses for all downstream tasks.
> > > >
> > > > As discussed in **Section 3.4**, large-scale outdoor scenes do not admit perfect analytic depth ground truth as in synthetic datasets. We therefore use high-accuracy LiDAR point clouds to generate dense depth maps, which serve as the reference geometry for evaluating depth-related tasks. Concretely, we first render RGB views from known LiDAR poses and treat these as anchor views in the LiDAR coordinate system. We then perform a joint aerial triangulation of the real RGB images and the LiDAR-rendered RGB views, while keeping the LiDAR anchor poses fixed. This solves all RGB camera poses consistently in the LiDAR frame. The aligned LiDAR point cloud is then projected into each RGB view to obtain pixel-wise dense depth maps. This pipeline directly supports the monocular depth estimation experiments reported in **Section 4.5** (see **Figure 6**), and the same mechanism can be used for event-based depth estimation.
> > > >
> > > > In large-scale UAV capture, pure visual SfM is prone to accumulated drift and local inconsistencies. In SkyEvents, LiDAR provides absolute geometric anchors: by locking LiDAR poses during the joint SfM optimization, we obtain globally consistent, drift-free 6-DoF poses for all RGB cameras in the LiDAR coordinate system. We hope this clarifies that LiDAR in SkyEvents is not just an additional modality, but a key enabler for reliable depth supervision, stable camera poses, and thus meaningful quantitative benchmarking for 3D reconstruction and related tasks.

---

> > > > > ### Author Response · Authors · 2025-12-03
> > > > > **Resonse to Weakness 5**
> > > > >
> > > > > We thank the reviewer for this insightful suggestion. In response, we have substantially enriched **Table 1**. In particular, we have added comparisons along additional dimensions such as RGB frame rate and sensor field-of-view (FoV) alignment between RGB and event cameras across different datasets. These new columns make it clearer that SkyEvents offers tightly aligned, overlapping FoVs and high-resolution, high-frame-rate RGB imagery, which are critical for reliable multi-view 3D reconstruction.
> > > > > Moreover, we have clarified and emphasized characteristics that are especially relevant for 3D tasks, including: whether a dataset provides globally consistent multi-view geometry, whether it includes LiDAR or dense depth supervision, and whether it is explicitly designed for 3D reconstruction under challenging conditions (low-light, motion blur, high dynamics).
> > > > >
> > > > > In the updated table and accompanying text, we highlight that SkyEvents is, to the best of our knowledge, **the first multi-view UAV event dataset explicitly designed for 3D reconstruction, with synchronized RGB–event–LiDAR, drift-free global poses, and dense depth derived from LiDAR**.

---

> > > > > > ### Author Response · Authors · 2025-12-03
> > > > > > **Resonse to Suggestions in the Future Work**
> > > > > >
> > > > > > ### a. In the Related Work section:
> > > > > > Thanks for your valuable advice, we have revised it
> > > > > > We thank the reviewer for this helpful suggestion regarding the structure of the Related Work section. We agree that the previous titles created unnecessary overlap. Accordingly, we have renamed Section 2.2 to "Event-based 3D Reconstruction" as suggested. This change significantly improves the clarity and organization of the literature review.
> > > > > >
> > > > > > ### b. Stereo event cameras:
> > > > > > We greatly appreciate the reviewer’s insightful suggestions regarding future research directions. We fully agree that employing stereo event cameras and exploring multi-UAV collaboration  would be significant steps forward for large-scale 3D scene reconstruction. We believe these advices point towards exciting avenues for future investigation.
> > > > > >
> > > > > > ### c. Cite more recent works:
> > > > > > We sincerely thank the reviewer for bringing this important work to our attention. We agree that citing recent advancements is crucial for a comprehensive literature review. Accordingly, we have added the citation for [1,2,3,4] in the Related Work section of the revised manuscript.
> > > > > >
> > > > > > [1] Active Event-based Stereo Vision, CVPR 2025.
> > > > > >
> > > > > > [2] Enhanced event-based dense stereo via cross-sensor knowledge distillation, ICCV 2025.
> > > > > >
> > > > > > [3] Physical-based event camera simulator, ECCV 2024.
> > > > > >
> > > > > > [4] EvaGaussians: Event stream assisted Gaussian splatting from blurry images, ICCV 2025.

---

> > > > > > > ### Author Response · Authors · 2025-12-03
> > > > > > > **Resonse to Question 1**
> > > > > > >
> > > > > > > We thank the reviewer for raising this important practical question. We address both the **spatial (mechanical) integration** and **temporal synchronization** below.
> > > > > > >
> > > > > > > To ensure that the event camera and frame-based camera experience the same motion and vibration, we designed a dedicated mechanical mount: (1) We first precisely measured the 3D dimensions of the EVK4 and DJI Action 4 and designed a custom carbon-fiber bracket that tightly houses both cameras. The cameras are press-fitted into the bracket so that their sides are in close frictional contact with the inner walls, effectively suppressing any relative micro-movements.
> > > > > > > (2) The two cameras are then rigidly coupled using metal screws that lock them into fixed positions on the bracket. This ensures that their optical axes are at the same height and their fields of view are largely overlapping, and that they move as a single rigid unit. Under UAV flight vibrations, both sensors thus experience nearly identical vibration frequency and amplitude.
> > > > > > > (3) Finally, the entire payload bracket is mounted to the UAV via a set of vibration-damping metal screw assemblies, which attenuate high-frequency vibrations transmitted from the UAV frame. This combination of rigid coupling between cameras and damped coupling to the UAV minimizes differential motion between the sensors.
> > > > > > >
> > > > > > > For time alignment, we use a shared control pipeline: (1) Both the DJI Action 4 and EVK4 are connected via data cables to the **same onboard computer**, which provides a unified system clock. (2) We implement a custom Python-based synchronization script that sends simultaneous start-acquisition commands to both cameras based on this common time base. (3) In our measurements, the average trigger skew between the two cameras is below 5 ms, which is further refined in our GTA alignment.
> > > > > > >
> > > > > > > To support reproducibility and adoption in the community, we plan to release the **CAD design files of the bracket (see APPENDIX)**, **wiring and power schematics**, and the synchronization control code together with the dataset.

---

> > > > > > > > ### Author Response · Authors · 2025-12-03
> > > > > > > > **Resonse to Question 2**
> > > > > > > >
> > > > > > > > Our UAV platform supports a wide range of flight speeds, with the maximum speed reaching up to **23 m/s**. For a conventional 20 FPS RGB camera, a typical shutter time in outdoor UAV flights is about 30 ms. At low altitude and slow speed, e.g., at 30 m altitude and 2 m/s horizontal velocity, the camera motion during one exposure can be kept below approximately one ground-sampling-distance (GSD), so motion blur is limited and the RGB frames remain reasonably sharp.
> > > > > > > >
> > > > > > > > However, as altitude and speed increase, the situation changes drastically. In our data collection, a large portion of the flights are conducted around **100 m** altitude, where the GSD is about **2.57 cm/pixel**. If we were to use a 20 FPS camera with a 30 ms exposure at higher UAV speeds, the camera would travel a non-negligible distance during a single exposure, leading to object displacements spanning many pixels in the image. In this regime, the 20 FPS RGB stream suffers from strong linear motion streaks and severely blurred textures, which directly harms feature matching and pose estimation for SfM and 3D reconstruction.
> > > > > > > >
> > > > > > > > If we relax these conservative constraints and allow the UAV to fly faster (recall our platform can go up to 23 m/s), then, at altitudes around 80–100 m, a 20 FPS RGB camera with 30 ms exposure would accumulate substantial motion during each frame, leading to pronounced motion blur and loss of high-frequency details across tens of pixels.
> > > > > > > >
> > > > > > > > In contrast, the event camera operates at microsecond temporal resolution and responds asynchronously to brightness changes. Even under fast motion and the same flight envelope, edges and contours in the event domain remain sharp, and the effective temporal sampling is far denser than 20 FPS. This is precisely the regime (higher speed, larger altitude, low light) where events provide a clear advantage over standard frame-based RGB.

---

### Meta-Review · Area_Chair_Ah8o · 2025-12-29

**Summary:**

The submission introduces SkyEvents, a large-scale multi-modal UAV dataset with RGB + event + LiDAR, designed for robust 3D reconstruction under challenging conditions (e.g., low light and motion blur). Reviewers who leaned positive emphasized the dataset’s scale/modality richness and the practical value of providing accurate geometry/poses and synchronization tooling. The main concerns driving discussion were (i) whether the paper provides enough task-level benchmarking and quantitative evidence beyond qualitative demonstrations, and (ii) whether auxiliary components (GTA alignment and RER loss) are sufficiently clear/novel, or should be positioned primarily as supporting tools for making the dataset usable.

Despite a split, I recommend acceptance because the dataset itself is substantial and timely (large-scale, multi-modal UAV data with geometry/poses), and the rebuttal materially improved clarity on synchronization/calibration and sharpened the framing that the dataset is the central contribution. The remaining concerns primarily affect the strength of auxiliary-method claims and the breadth of benchmarking, but do not outweigh the community value of releasing this resource and baseline pipeline.

**Reviewer Concerns:**

- **Task-level benchmarking depth (raised by WeKv):** WeKv consistently emphasized that a dataset paper would benefit from stronger task-specific quantitative evidence and clearer benchmarking protocols. While the rebuttal improved several reproducibility and alignment details, this remains a reasonable request that could further strengthen the paper’s impact as a benchmark resource.

- **Clarity/positioning of auxiliary components (raised by r6JB):** r6JB provided detailed feedback that GTA and especially RER should be presented with clearer definitions, intuition, and cleaner evidence of benefit, and noted that some performance gains from events are not uniformly consistent across settings. Even if the authors position GTA/RER primarily as supporting tools, incorporating these clarifications would make the paper easier to evaluate and reuse.

**Reviewer Scores:**

- **jye9:** 6 → 6 (up to 8). Already marginally above threshold.

- **3NrW:** 6 → 6. Supportive overall with concrete requests; likely to remain just above threshold.

- **WeKv:** While the current score is low, the concerns focus on the depth of task-level quantitative benchmarking and reproducibility. If the final version further strengthens benchmarking protocols and quantitative evaluations, an upward revision is plausible; however, based on the current discussion record, I do not assume a score change.

- **r6JB:** The score reflects persistent concerns about the clarity/positioning of GTA/RER and aspects of the evaluation structure. With clearer definitions, stronger evidence of benefit, and a more consistent presentation of results, the score could plausibly move upward, though the current record does not guarantee that.

---

### Decision · Program_Chairs · 2026-01-26

Accept (Poster)